# Role of posterior medial thalamus in the modulation of striatal circuitry and choice behavior

Alex J Yonk, Ivan Linares-García, Logan Pasternak, Sofia E Juliani, Mark A Gradwell, Arlene J George, David J Margolis*

Department of Cell Biology and Neuroscience, Rutgers, The State University of New Jersey, Piscataway, United States

## eLife Assessment

Yonk and colleagues provide a **valuable**, timely, and in-depth study showcasing the role of thalamostriatal inputs in learning and action selection. After characterizing the synaptic properties of these inputs onto different striatal cell types in vitro, they provide **solid** evidence that posterior medial thalamic nucleus (POm) terminals in striatum are activated during reward expectation and arousal. The overall function of this pathway and the degree to which results are confounded by viral contamination of surrounding nuclei and movements remain open questions.

*For correspondence:
david.margolis@rutgers.edu

Competing interest: The authors declare that no competing interests exist.

**Abstract** The posterior medial (POm) thalamus is heavily interconnected with sensory and motor circuitry and is likely involved in behavioral modulation and sensorimotor integration. POm provides axonal projections to the dorsal striatum, a hotspot of sensorimotor processing, yet the role of POm-striatal projections has remained undetermined. Using optogenetics with mouse brain slice electrophysiology, we found that POm provides robust synaptic input to direct and indirect pathway striatal spiny projection neurons (D1- and D2-SPNs, respectively) and parvalbumin-expressing fast spiking interneurons (PVs). During the performance of a whisker-based tactile discrimination task in head-restrained mice, POm-striatal projections displayed learning-related activation correlating with anticipatory, but not reward-related, pupil dilation. Inhibition of POm-striatal axons across learning caused slower reaction times and an increase in the number of training sessions for expert performance. Our data indicate that POm-striatal inputs provide a behaviorally relevant arousal-related signal, which may prime striatal circuitry for efficient integration of subsequent choice-related inputs.

## Introduction

The process of sensorimotor learning is underpinned by sensory perception and motor control (*Petersen, 2019*). In the mouse whisker system, tactile sensations are acquired via active sensor (e.g. whisker) movement to obtain relevant environmental information and subsequent processing by the well-characterized primary somatosensory barrel cortex (S1) circuitry (*Petersen, 2019*; *El-Boustani et al., 2020*). This whisker-related information is transmitted from the periphery to S1 via two thalamic nuclei, ventral posterior medial (VPM) and posterior medial (POm), constituting the lemniscal and paralemniscal pathways, respectively (*Deschênes et al., 2003*; *Bureau et al., 2006*; *Petersen, 2007*; *Diamond et al., 2008*; *Moore et al., 2015*; *Deschenes and Urbain, 2016*; *Yu et al., 2006*; *Mo et al., 2017*). VPM reliably encodes fast-whisking components including self-motion and tactile information (*Moore et al., 2015*; *Chiaia et al., 1991b*; *Diamond et al., 1992*; *Sosnik et al., 2001*; *Urbain et al., 2015*). Conversely, POm encodes phase-related whisking activity with relatively lower magnitude

responses and higher response failure rates (*Moore et al., 2015*; *Diamond et al., 1992*; *Sosnik et al., 2001*; *Urbain et al., 2015*; *Chiaia et al., 1991a*; *Lavallée et al., 2005*; *Masri et al., 2008*; *Petty et al., 2021*). Recent work has highlighted two behavior-related aspects of POm function: (1) activation during changes in behavioral state, especially related to sensory and nociceptive processing (*Moore et al., 2015*; *Chiaia et al., 1991b*; *Diamond et al., 1992*; *Sosnik et al., 2001*; *Masri et al., 2008*; *Petty et al., 2021*; *Noseda et al., 2010*; *Frangeul et al., 2014*; *Osaki et al., 2022*) and (2) driving learning-related plasticity at its cortical synapses (*Gambino et al., 2014*; *Audette et al., 2019*; *Williams and Holtmaat, 2019*; *Zhang and Bruno, 2019*; *La Terra et al., 2022*).

POm receives a plethora of inputs including glutamatergic (S1, primary motor cortex (M1), secondary somatosensory cortex, superior colliculus, and spinal trigeminal interpolaris; *Chiaia et al., 1991a*; *Roger and Cadusseau, 1984*; *Pierret et al., 2000*; *Alloway et al., 2003*; *Alloway et al., 2008*; *Groh et al., 2014*; *Yamawaki and Shepherd, 2015*; *Sumser et al., 2017*; *Gharaei et al., 2020*), GABAergic (ventral zona incerta, anterior pretectal nucleus, and thalamic reticular nucleus; *Lavallée et al., 2005*; *Roger and Cadusseau, 1985*; *Power et al., 1999*; *Barthó et al., 2002*; *Power and Mitrofanis, 2002*; *Trageser and Keller, 2004*; *Barthó et al., 2007*; *Giber et al., 2008*; *Watson et al., 2015*), and cholinergic (pedunculopontine and laterodorsal tegmental nuclei; *Masri et al., 2006*; *Trageser et al., 2006*; *Park et al., 2017*; *Huerta-Ocampo et al., 2020*). Further, the stereotypical POm-cortical projection terminates in S1, specifically layers (L)1 and L5A (*Bureau et al., 2006*; *Meyer et al., 2010*; *Wimmer et al., 2010*; *Ohno et al., 2012*) and has been studied in the context of driving cortical plasticity/perceptual learning (*Gambino et al., 2014*; *Audette et al., 2019*; *Williams and Holtmaat, 2019*; *Zhang and Bruno, 2019*; *La Terra et al., 2022*; *Audette et al., 2018*; *Qi et al., 2022*). In addition to its cortical projection, POm axons pass through and collateralize with terminal synaptic boutons in both thalamic reticular nucleus and posterior dorsolateral striatum (pDLS) as they ascend toward cortex (*Ohno et al., 2012*; *Spreafico et al., 1987*; *Descheˆnes et al., 1995*; *Viaene et al., 2011*; *Smith et al., 2012*; *Alloway et al., 2017*; *Li et al., 2020*; *O'Reilly et al., 2021*). Here, we focus on the POm-striatal projection as striatal circuitry modulation may have powerful effects on sensorimotor integration and behavior (*Yonk and Margolis, 2019*). However, POm's influence over striatal microcircuitry and behavioral performance is unresolved.

The striatum is the predominant input nucleus of the basal ganglia and is predominantly composed of GABAergic spiny projection neurons (SPNs) expressing either D1 or D2 dopamine receptors (*Gerfen et al., 1996*; *Gerfen and Surmeier, 2011*; *Tritsch and Sabatini, 2012*), but it also contains a rich diversity of interneurons, such as parvalbumin-expressing (PV) fast-spiking interneurons that exert robust modulatory control over SPN output (*Gittis et al., 2010*; *Tepper et al., 2010*; *Lee et al., 2017*; *Owen et al., 2018*; *Tepper et al., 2018*). Within this microcircuitry, the dorsal striatum integrates widespread convergent cortical and thalamic inputs that constitute part of the force driving normal striatal functioning (*Pan et al., 2010*; *Wall et al., 2013*; *Huerta-Ocampo et al., 2014*; *Oh et al., 2014*; *Guo et al., 2015*; *Hunnicutt et al., 2016*; *Hintiryan et al., 2016*; *Smith et al., 2016*; *Hooks et al., 2018*). Notably, functionally related cortical (S1 and M1) and thalamic (POm) inputs converge within shared striatal subregions (*Hunnicutt et al., 2016*; *Hintiryan et al., 2016*; *Alexander et al., 1986*). For example, M1 and S1 are heavily interconnected via reciprocal L2/3 and L5 connections, (*Mao et al., 2011*; *Chen et al., 2013a*; *Kwon et al., 2016*). and their projections overlap within dorsal striatum and even onto the same neuron (*Hunnicutt et al., 2016*; *Alloway et al., 2000*; *Hoffer and Alloway, 2001*; *Ramanathan et al., 2002*; *Charpier et al., 2020*; *Smith et al., 2022*; *Sanabria et al., 2024*). While some studies treat striatal inputs as a uniform entity (*Ding et al., 2008*; *Doig et al., 2014*), they have been shown to differ anatomically (*Hunnicutt et al., 2016*; *Hintiryan et al., 2016*; *Hooks et al., 2018*), functionally (*Lee et al., 2019*; *Johansson and Silberberg, 2020*) and behaviorally (*Lee et al., 2019*; *Sun et al., 2021*; *Zareian et al., 2023*). Thus, the specific cortical and thalamic origin of striatal inputs likely has significance for understanding how the striatal circuitry integrates sensorimotor information to modulate behavior.

The most prominent thalamostriatal modulation occurs via parafascicular (Pf) thalamus (*Johansson and Silberberg, 2020*; *Smith and Parent, 1986*; *Berendse and Groenewegen, 1990*; *Lapper and Bolam, 1992*; *Matsumoto et al., 2001*; *Ellender et al., 2013*; *Parker et al., 2016*; *Mandelbaum et al., 2019*; *Tanimura et al., 2019*; *Fallon et al., 2023*). Pf is implicated in regulating action flexibility (*Brown et al., 2010*; *Bradfield et al., 2013*) and contributing to the initiation and execution of learned sequences of movements (*Fallon et al., 2023*; *Díaz-Hernández et al., 2018*) through its

robust innervation of striatal cholinergic interneurons (*Lapper and Bolam, 1992*; *Ding et al., 2008*; *Doig et al., 2014*; *Johansson and Silberberg, 2020*). Conversely, despite direct comparisons to Pf, (*Alloway et al., 2017*) POm's functional innervation pattern and subsequent influence over the striatal microcircuitry and choice behavior is undetermined (*Yonk and Margolis, 2019*). Here, we used ex vivo whole-cell recordings of identified D1-SPNs, D2-SPNs, and PV interneurons to assess the functional connectivity of POm-striatal projections, and in vivo fiber photometry and photoinactivation to identify the contribution of POm-striatal axonal activity on sensory-guided behavioral performance and learning.

## Results

### POm equally innervates striatal cell types with faster latency in PV interneurons

We stereotaxically injected pAAV-ChR2-EYFP unilaterally in POm, permitting channelrhodopsin (ChR2) expression in its thalamostriatal terminals to investigate the relative synaptic strength of POm inputs onto three identified striatal neurons (D1-SPNs, D2-SPNs, and PV interneurons; *Figure 1A–B*). The injection site was confirmed by verifying the stereotypical POm-cortical projection pattern (S1 L1 and L5a; *Figure 1C*; *Bureau et al., 2006*; *Zhang and Bruno, 2019*; *La Terra et al., 2022*; *Meyer et al., 2010*; *Ohno et al., 2012*). In acute ex vivo brain slices, neurons were targeted for patch clamp recordings within pDLS (AP from bregma = –0.34––1.22) corresponding with the POm-striatal axonal projection field (*Figure 1—figure supplement 1*; *Alloway et al., 2017*; *Alloway et al., 2014*). Striatal neurons were identified and targeted by crossing their respective Cre-recombinase mouse lines with tdTomato-expressing reporter mice and validating their intrinsic electrophysiological properties in response to hyperpolarizing and depolarizing current steps (*Figure 1—figure supplement 1A–D, G-L*; see Materials and methods; *Tepper et al., 2010*; *Tepper et al., 2018*; *Kawaguchi, 1993*). Whole-cell current-clamp recordings were performed without inhibitory synaptic blockers to resemble natural physiological responses to optogenetic activation of POm inputs (*Lee et al., 2019*). After breaking in, patched cells were subjected to a standard set of protocols: (1) hyperpolarizing and depolarizing current steps to define intrinsic firing properties and optogenetic; and (2) single pulse (SP), (3) paired-pulse ratio (PPR), and (4) train stimulation to measure synaptic responses.

Optogenetic activation of POm terminals readily elicited depolarizing postsynaptic potentials (PSP) in all targeted cell types (*Figure 1D*). Responses to SP stimulation resulted in relatively equal PSP amplitudes for D1-SPNs (7.05±0.75 mV), D2-SPNs (8.79±1.67 mV), PVs (10.83±1.91 mV), and neighboring unlabeled cells that we termed putative SPNs, based on their intrinsic firing properties (5.54±1.04 mV) ($F_{(4,51)}$ = 2.455, p=0.4835; *Figure 1E*). Three PV interneurons and one D1-SPN exhibited action potentials to SP stimulation and were excluded from further analysis. A small but significant correlation was observed between PSP amplitude and increasing distance from the injection site ($r^2$=0.08493, p=0.0293, n=59 cells; *Figure 1—figure supplement 1E–F*). The latency to maximum PSP amplitude was significantly shorter in PVs (7.29±0.32ms, n=17 cells from 7 mice) than D1-SPNs (11.05±0.45ms, n=20 cells from 6 mice), D2-SPNs (10.68±0.56ms, n=11 cells from 5 mice), and putative SPNs (10.81±0.92ms, n=7 cells from 4 mice; $F_{(4,51)}$ = 29.78, p<0.0001, PV vs. D1 p<0.0001, PV vs. D2 p=0.0003, PV vs. SPN, p=0.0056; *Figure 1F*). In a subset of recordings, identified and unidentified cells within the same field of view on the same slice were patched sequentially to control for injection site variability. D1- and D2-SPNs did not differ in PSP amplitude (9.55±2.83 mV in D1-SPNs vs. 8.48±2.15 mV in D2-SPNs, p=0.6406, n=8 pairs, N=5) or latency (10.13±0.66ms in D1-SPNs vs. 10.92±0.56ms in D2-SPNs, p=0.3254, n=8 pairs, N=5; *Figure 1—figure supplement 1M–O*). In contrast, sequentially patched PV and SPNs did not significantly differ in maximum PSP amplitude (10.26±3.05 mV in PVs vs. 4.76±0.97 mV in SPNs, p=0.2500, n=9 pairs, N=7), but PVs had faster latency (6.85±0.42ms in PVs vs. 10.75±0.87ms in SPNs, p=0.0006, n=9 pairs, N=7; *Figure 1—figure supplement 1P–R*), validating the population results. Strikingly, we found relatively equal PSP amplitudes in all recorded cell types within pDLS, indicating that POm provides robust and unbiased synaptic input to all targeted striatal cells.

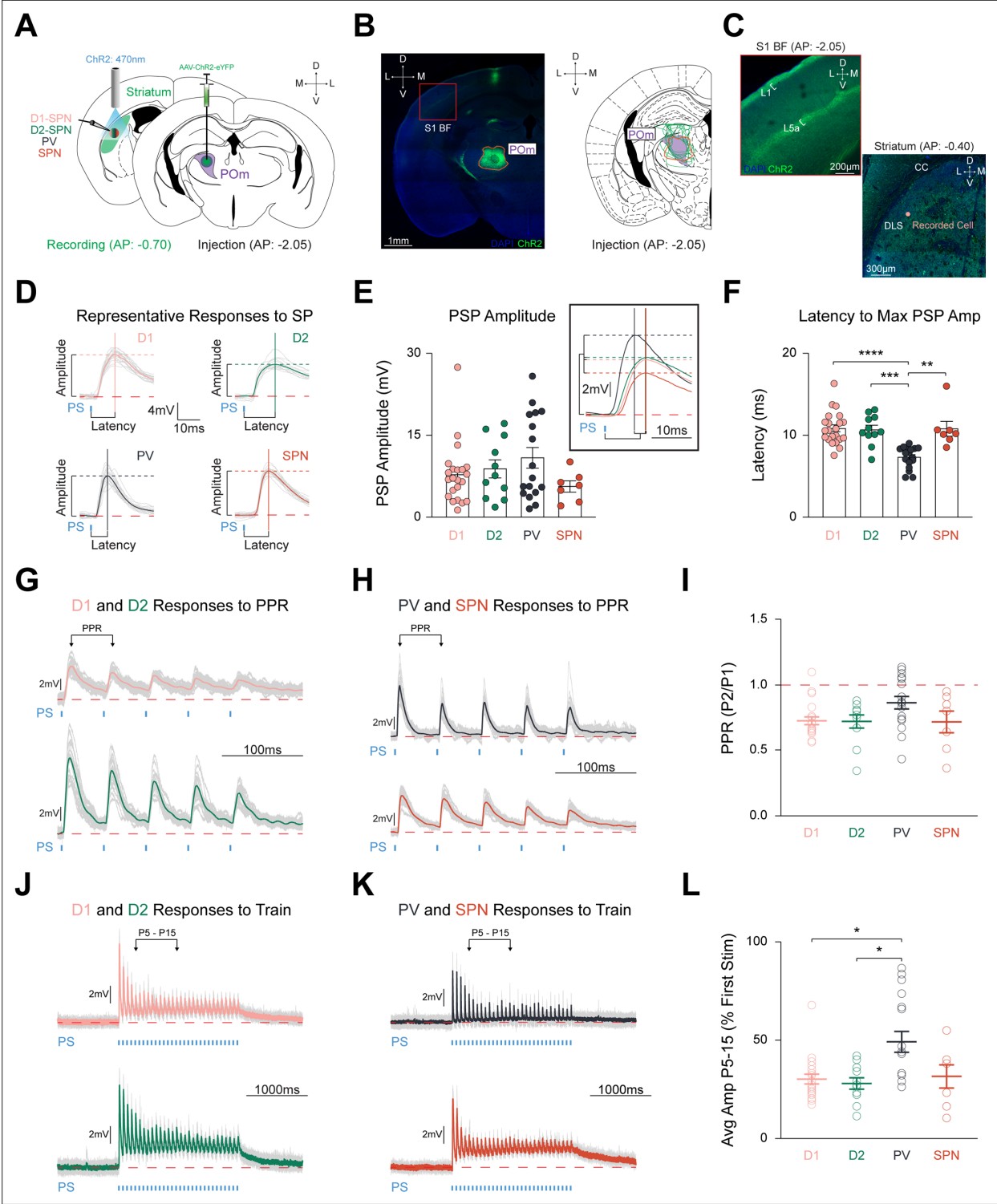

**Figure 1.** POm equally innervates striatal cell types with faster latency in PV interneurons. (**A**) Schematic detailing pAAV-ChR2-EYFP injection unilaterally into POm (*Right*), and optogenetic stimulation of POm-striatal afferents whilst recording from identified and unidentified neurons via ex vivo slice of posterior DLS (AP range: –0.34 to –1.22 relative to Bregma; *Left*). See *Figure 1—figure supplement 1*. Illumination (2.5ms pulses of 470 nm light,~0.6 mW intensity) was delivered through the 40 x objective. (**B**) Representative injection site (orange) in POm (*Left*), and viral spread of all electrophysiology injections within highlighted POm (purple; *Right*). S1BF = S1 Barrel Field. Scale = 1 mm. (**C**) Red box inset from panel (**B**) highlighting stereotypical POm-cortical projection pattern to S1BF L1 and L5a (*Zhang and Bruno, 2019*; *La Terra et al., 2022*; *Ohno et al., 2012*). *Right*: POm-striatal axons within posterior DLS. CC = corpus callosum. Scale = 200 μm. (**D**) Representative cell type-specific PSPs to SP stimulation. Colored lines

*Figure 1 continued on next page*

*Figure 1 continued*

= average PSP of 20 sweeps. Gray lines = 20 individual traces. Solid vertical and dashed horizontal lines = latency and amplitude, respectively. Red dashed line = 0 mV. Blue tick = photostimulation (PS). Time scale = 10ms. Voltage scale = 4 mV. (**E**) Amplitudes evoked by each cell type were similar (D1-SPNs=20 cells from 6 mice, D2-SPNs=11 cells from 5 mice, PVs = 17 cells from 7 mice, unidentified SPNs = 7 cells from 4 mice). Inset shows grand average PSPs. Time scale = 10ms. Voltage scale = 2 mV. (**F**) Latency to maximum PSP amplitude is significantly quicker in PVs than all other cell types. (**G–H**) Representative responses of (**G**) D1-SPN (*Top*) and D2-SPN (*Bottom*), and (**H**) PV (*Top*) and putative SPN (*Bottom*) to PPR stimulation (*Assous and Tepper, 2019*). PPR is defined as the ratio of PSP amplitude of pulse 2 over the ratio of PSP amplitude of pulse 1. PPR PS parameters = five 2.5ms pulses with 50ms interpulse intervals (20 Hz). Time scale = 100ms. Voltage scale = 2 mV. See *Figure 1—figure supplement 2I* Stimulation of POm-striatal afferents evokes similar PPR responses. (**J–K**) Representative responses of (**J**) D1-SPN (*Top*) and D2-SPN (*Bottom*), and (**K**) PV (*Top*) and putative SPN (*Bottom*) to train stimulation (*Landisman and Connors, 2007*). Colored lines = average of five individual gray traces. Train PS parameters = thirty 2.5ms pulses with 64.2ms interpulse intervals (15 Hz). Time scale = 1000ms. Voltage scale = 2 mV. (**L**) Relative PSP amplitude (average of pulses 5–15 compared to pulse 1) is significantly larger than both SPNs. Data are mean ± SEM. *p<0.05, ** p<0.01, *** p<0.001, **** p<0.0001.

The online version of this article includes the following figure supplement(s) for figure 1:

**Figure supplement 1.** Recorded cell location, intrinsic electrophysiological parameters, and sequentially patched PSP amplitude and latency.

**Figure supplement 2.** Representative responses of identified striatal cells to PPR and train stimulation.

## Short-term synaptic dynamics are similar between striatal cell types, but synaptic depression is milder in PV interneurons

The strength of synaptic inputs varies dramatically based on activation frequency and in a cell-type-specific manner with robust synaptic contacts generally exhibiting synaptic depression (*Lee et al., 2019*; *Johansson and Silberberg, 2020*; *Abbott et al., 1997*; *Zucker and Regehr, 2002*). To fully characterize the relative synaptic strength of POm-striatal inputs in a cell-type-specific manner, we assessed short-term plasticity by applying a PPR protocol of five pulses (*Figure 1G–H*, *Figure 1—figure supplement 2A–B*; *Assous and Tepper, 2019*; *Glasgow et al., 2019*). While all cell types exhibited robust synaptic depression overall, no PPR differences were observed (D1-SPNs: 0.73±0.03, n=20, D2-SPNs: 0.72±0.05, n=11, PVs: 0.86±0.05, n=17, SPNs: 0.71±0.08, n=7; $F_{(4,51)}$ = 7.101, p=0.0688; *Figure 1I*).

To further characterize short-term synaptic dynamics, we applied a train protocol of thirty pulses at a frequency characteristic of POm-striatal activity (*Figure 1J–K*; *Petty et al., 2021*; *Landisman and Connors, 2007*). Similar to PPR stimulation, train stimulation elicited overall synaptic depression in all cell types, but PV interneurons exhibited a milder synaptic depression relative to both SPN types that occurred predominantly between pulses 5–15 (*Figure 1L*, *Figure 1—figure supplement 2*). PV interneurons (0.49±0.05, n=17) showed significant differences compared to D1-SPNs (0.30±0.03, n=20) and D2-SPNs (0.28±0.03, n=11), but not with SPNs (0.31±0.06; $F_{(4,51)}$ = 10.99, p=0.0118, D1-SPN vs. PV p=0.0184, D2-SPN vs. PV p=0.0481, PV vs. SPN p=0.4940; *Figure 1L*). Thus, POm-striatal projections provide SPNs and PV interneurons with unbiased and large amplitude synaptic inputs, characterized by milder synaptic depression in PVs, highlighting a potentially significant role in modulating striatal microcircuitry (*Lee et al., 2019*; *Johansson and Silberberg, 2020*).

## Mice rapidly learn to discriminate between two textures

While specific sensorimotor integrative and learning roles have been proposed and tested for several striatal inputs (*Wall et al., 2013*; *Guo et al., 2015*; *Ding et al., 2008*; *Lee et al., 2019*; *Johansson and Silberberg, 2020*; *Díaz-Hernández et al., 2018*; *Kawai et al., 2015*; *Rothwell et al., 2015*; *Kupferschmidt et al., 2017*), the role of POm-striatal projections is still unknown. To monitor the activation of POm-striatal projections, we injected pAAV-axon-jGCaMP8s unilaterally into left POm and implanted a 400 μm core cannula into left pDLS (*Figure 2A–B*). Violet light (405 nm) and blue light (470 nm) were constantly delivered to pDLS throughout the entire session to measure the isosbestic and POm-pDLS axonal calcium signals, respectively. Using a similar protocol from our previous publication (*Lee et al., 2019*), water-restricted wild-type mice were trained on a whisker-based discrimination (Go/NoGo) paradigm. Mice received water for licking correctly (Hit) to the Go texture (P100 sandpaper). They received a white noise tone and a 12 s time-out period for licking incorrectly (False Alarm; FA) to the NoGo texture (P1200 sandpaper; *Figure 2C*). If mice did not lick to the Go or NoGo texture, the texture retreated to its starting point, and trials were considered Miss and Correct Rejection (CR), respectively (*Figure 2D*). Additionally, pupil dynamics are a known metric of arousal (*Kahneman and Beatty, 1966*; *Murphy et al., 2014*), correlate well with POm activity (*Petty et al.,*

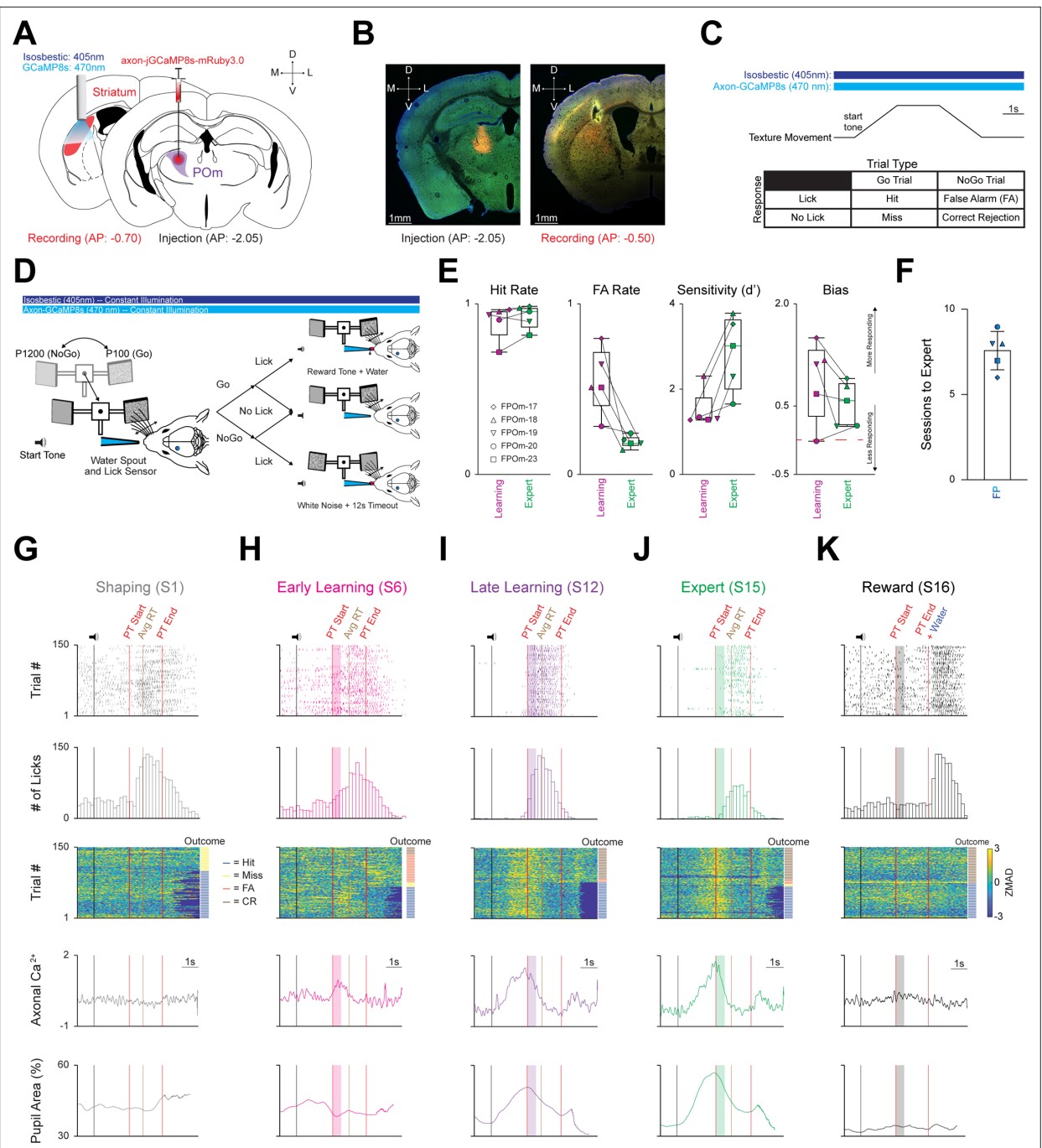

**Figure 2.** Mice rapidly learn to discriminate between two textures, and all three activity parameters markedly increase across learning. (**A**) Schematic detailing pAAV-hSynapsin1-axon-jGCaMP8s-P2A-mRuby3 injection unilaterally into POm (*Right*), and a 400 μm cannula implanted in the left posterior DLS (*Left*). (**B**) Representative injection site in POm (*Left*), and cannula placement in the posterior DLS along with ascending POm axons (*Right*). Scale = 1 mm. (**C**) *Top*: Stimulating timing and texture movement representation during a trial. Note = both LEDs (isosbestic = 405 nm; axon-jGCaMP8s=470 nm) were constantly on for every session. *Bottom*: Outcomes for each stimulus-response pair. (**D**) Schematic representing texture movement and potential outcomes during a single trial of the Go/NoGo whisker discrimination paradigm. (**E**) Changes in Hit Rate, FA Rate, Sensitivity (**d'**) and Bias of the FP cohort (n=5 mice) as they transition from the Learning to the Expert phase. Note that mice are classified as Expert when they achieve a Hit Rate ≥0.80 and a FA Rate ≤0.30 for two consecutive sessions. Red line = 0. See *Figure 2—figure supplement 1*. (**F**) Average number of sessions required for expert discrimination of the FP cohort. (**G–K**) Three activity parameters (licking, axonal calcium, and pupil activity) from a representative (**G**) Shaping (session 1), (**H**) Early Learning (first two sessions after Shaping), (**I**) Late Learning (last two sessions before Expert), (**J**) Expert, and (**K**) Reward sessions from the same mouse (FPOm-18). *Top*: licking activity within a session (150 trials). Colored ticks = lick. Vertical black line = sound cue representing trial start as the texture moves toward the whisker field. Vertical red lines = start (texture arrival at endpoint in whisker field) and end (texture departure

*Figure 2 continued on next page*

**Figure 2 continued**

toward starting point) of the PT window (time where mice can respond by licking). Vertical brown line = average reaction time (RT; time of first lick that triggers an outcome) across all trials in each session. Colored boxes = 500ms grace period (licking does not trigger any outcomes). Note = no response line is present in the Reward session (**K**) as licking does not trigger any outcomes, and water was automatically delivered at PT end. *Top Middle*: Lick histogram. *Middle*: Heatmap sorted by trial outcome (to the *Right* of heatmap) highlighting axonal ZMAD calcium activity for each trial. Trial outcome is color coded (blue = Hit, yellow = Miss, orange = FA, brown = CR). *Bottom Middle*: Average axonal calcium activity of 150 trials for each session. *Bottom*: average pupil area (as a percentage) of 150 trials for each session. Data are mean ± SEM. Time scale = 1 s.

The online version of this article includes the following figure supplement(s) for figure 2:

**Figure supplement 1.** Individual longitudinal learning-related changes in behavioral parameters, licking activity, and calcium activity, and methodology of measuring pupil dynamics.

---

*2021*; *La Terra et al., 2022*), and exhibit outcome-dependent differences during the Go/NoGo paradigm (*Lee and Margolis, 2016*). Therefore, synchronized orofacial video was captured during behavioral performance, and deep-learning (*Mathis et al., 2018*; *Nath et al., 2019*) pupillometry (*Yamada and Toda, 2022*) was applied to assess pupil dynamics during task performance (*Figure 2—figure supplement 1B*) in which our results mirrored previously reported outcome-dependent differences (*Lee and Margolis, 2016*).

Mice in the fiber photometry (FP) cohort (n=5) underwent three training phases (Shaping, Learning, and Expert; *Figure 2—figure supplement 1A*) that were segmented into five discrete behavioral time points (Shaping, Early Learning, Late Learning, Expert, and Reward; see Materials and methods). During the learning phase, Hit rate increased, and FA rate decreased significantly, leading to markedly increased sensitivity (d') and slightly decreased bias (*Figure 2E*, **S3A**). Mice were considered Expert once they had reached ≥0.80 Hit Rate and ≤0.30 FA Rate for two consecutive sessions in lieu of a strict sensitivity (d') threshold; we found this definition more intuitive because d' is enhanced as Hit Rate and FA Rate approach their extremes (0 or 1; *Figure 2—figure supplement 1A*). On average, it took this cohort 7.6±0.51 sessions to become Experts (*Figure 2F*). Thus, the FP cohort rapidly learned to discriminate between the two textures as validated by behavioral responding parameters.

## Calcium activity markedly increases and becomes stereotyped across learning

To elucidate the contribution of POm-striatal projections during the Go/NoGo discrimination task, we measured POm axonal calcium activity along with licking and pupil activity at five discrete behavioral time points (Shaping, Early Learning, Late Learning, Expert, and Reward; *Figure 2G–K*). Photometry measurements revealed learning-related increases in POm axonal activity, starting before and peaking near texture presentation (*Figure 3A*, *Figure 3—figure supplement 1A*) as measured by two parameters: calcium signal amplitude (ZMAD; median absolute deviation of the z-score) and area under the curve of the receiver-operator characteristic (auROC). First, the average maximum calcium amplitude significantly increased across learning ($F_{(1.818,7.271)}$=39.24, p=0.0001, n=5; *Figure 3B*). Further, the average maximum amplitude at Shaping (0.27±0.04) was significantly smaller compared to Early Learning (1.04±0.12, p=0.0190), Late Learning (1.58±0.18, p=0.0061), and Expert (1.59±0.17, p=0.0052). To test whether POm activity was reward-related, a single session was performed following the Expert phase during which the textures were removed, and water was automatically provided at the end of the presentation time (PT) window. We observed that calcium activity regressed to Shaping levels and was significantly smaller than Early (p=0.0306), Late (p=0.0126), and Expert (p=0.0098).

Second, to quantify POm axonal activity within the striatum relative to learning, we employed signal detection theory, utilizing auROC values to compare the basal activity across the five behavioral time points as learning occurs (*Li et al., 2017*; *Kingsbury et al., 2019*). As with the maximum calcium amplitude, auROC values significantly increased across learning ($F_{(2.162,8.650)}$=51.17, p<0.0001, n=5; *Figure 3C*). Notably, auROC values at Shaping (0.54±0.01) were significantly lower than Early Learning (0.70±0.03, p=0.0116), Late Learning (0.83±0.03, p=0.0017), and Expert (0.85±0.02, p=0.0007), but not Reward (0.57±0.01, p=0.4111). Further, significant auROC differences were also noted between the Early Learning vs. Expert (p=0.0383), Late vs. Reward (p=0.0084), and Expert vs. Reward (p=0.0042). Finally, calcium-related events increased longitudinally and became stereotyped within a 4 s target window (centered around texture presentation) compared to a non-task-related 1 s control window (during the 1 s pre-task interval prior to the sound cue and texture movement;

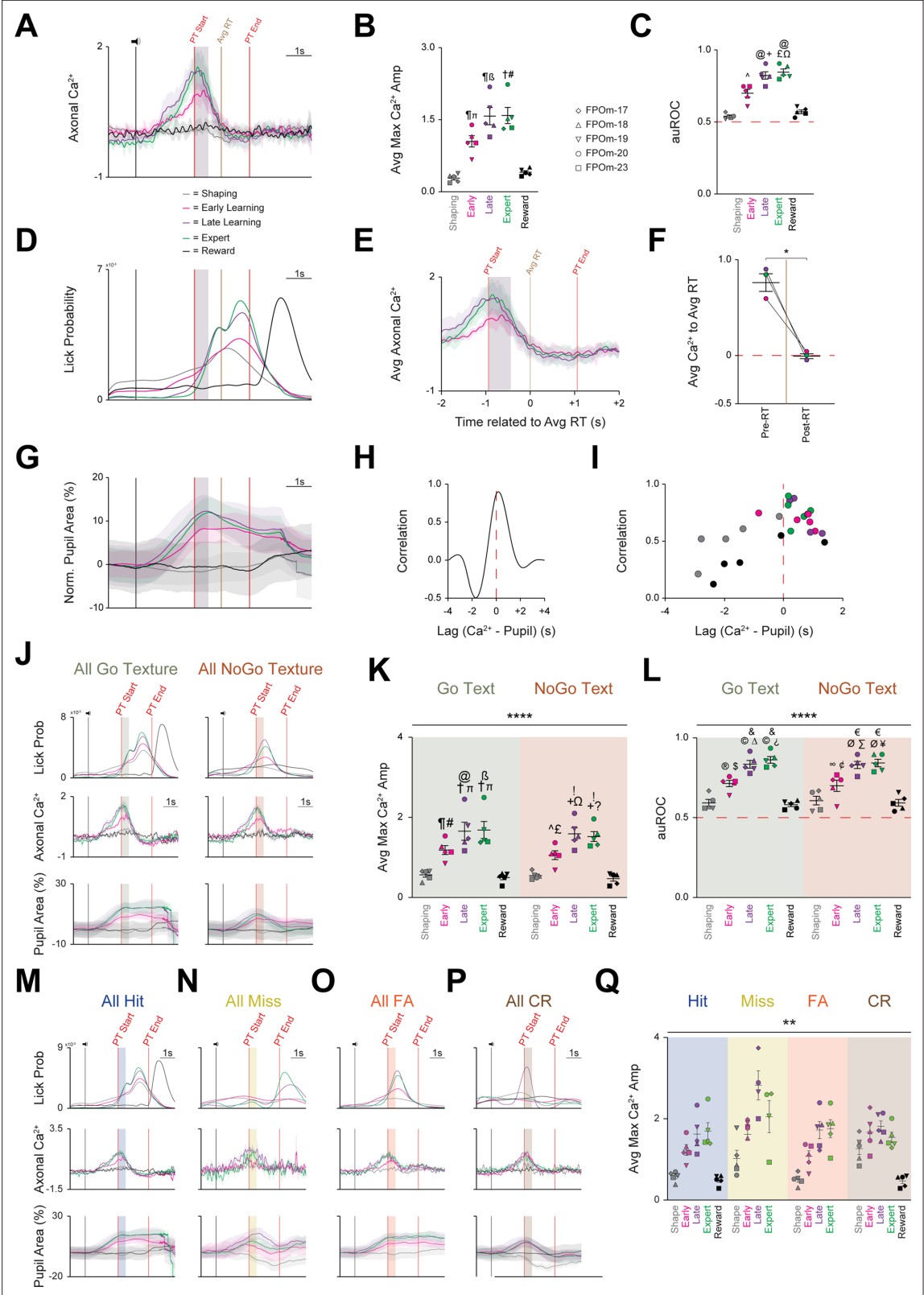

**Figure 3.** All three activity parameters exhibit marked increases across learning, but only axonal calcium activity remains unchanged, irrespective of trial type or outcome segmentation. (**A**) Grand average axonal calcium activity at each behavioral time point. Data are mean ± SD. (**B**) Average of maximal axonal calcium amplitude markedly increases across learning before regressing to Shaping levels during the Reward session. π p<0.01 Shaping vs. Early, ß p<0.001 Shaping vs. Late, # p<0.001 Shaping vs. Expert, ¶ p<0.05 Early/Late vs. Reward, † p<0.001 Expert vs. Reward. (**C**) Average area under the

*Figure 3 continued on next page*

Figure 3 continued

curve of the receiver-operator characteristic (auROC) also markedly increases across learning before regressing to Shaping levels during the Reward session. ^ p<0.05, Shaping vs. Early;+p < 0.01, Shaping vs. Late; £ p<0.001, Shaping vs. Expert; Ω p<0.05, Early vs. Expert; @ p<0.01, Late/Expert vs. Reward. (**D**) Grand average probability density function for licking-related activity at each time point. (**E**) Axonal calcium activity at Learning and Expert time points 2 s pre and 2 s post grand average RT. Data are mean ± SD. (**F**) Pre-RT axonal calcium activity is significantly larger than post-RT axonal calcium activity. (**G**) Grand average of normalized pupil area at each behavioral time point. (**H**) Representative cross-correlation of pupil area and axonal calcium activityI Cross-correlation of pupil area and calcium activity plotted for each behavioral time point for each mouse. (**J**) Grand average of all licking (*Top*), calcium (*Middle*), and normalized pupil (*Bottom*) activity segmented by trial type: Go texture (*Left*) and NoGo texture (*Right*) presentation. (**K**) Average of maximal axonal calcium amplitude markedly increases across learning for both Go and NoGo texture presentation before regressing to Shaping levels during the Reward session. ¶ p=0.0016 Go Shaping vs. Go Early, π p<0.0001 Go Shaping vs. Go Late/Expert, @ p=0.0265 Go Early vs. Go Late, ß p=0.0154 Go Early vs. Go Expert, # p=0.0005 Go Early vs. Go Reward, † p<0.0001 Go Late/Expert vs. Go Reward. ^ p=0.0128 NoGo Shape vs. NoGo Early,+p<0.0001, NoGo Shape vs. NoGo Late/Expert, £ p=0.0031 NoGo Early vs. NoGo Reward, Ω p=0.0085 NoGo Early vs. NoGo Late,? p=0.0295 NoGo Early vs. NoGo Expert,! p<0.0001, NoGo Late/Expert vs. NoGo Reward. (**L**) Average auROC markedly increases across learning for both Go and NoGo texture presentation before regressing to Shaping levels during the Reward session. p=0.0041 Go Shaping vs. Go Early, p<0.0001 Go Shaping vs. Go Late/Expert, Δ p=0.0041 Go Early vs. Go Late, > p=0.0003 Go Early vs. Go Expert, $ p=0.0014 Go Early vs. Go Reward, and p<0.0001 Go Late/Expert vs. Go Reward. ∞ p=0.0462 NoGo Shaping vs. NoGo Early, Ø p<0.0001 NoGo Shaping vs. NoGo Late/Expert, ∑ p=0.0017 NoGo Early vs. NoGo Late, ¥ p=0.0005 NoGo Early vs. NoGo Expert, ¢ p=0.0142 NoGo Early vs. NoGo Reward, € p<0.0001 NoGo Late/Expert vs. NoGo Reward. (**M–P**) Grand average of licking (*Top*), calcium (*Middle*), and normalized pupil (*Bottom*) activity segmented by trial outcomes: (**M**) Hit, (**N**) Miss, (**O**) FA, and (**P**) CR. (**Q**) Average of maximal axonal calcium amplitude of each mouse markedly increases across learning for all trials outcomes before regressing to Shaping levels during the Reward session. Data are mean ± SEM unless noted otherwise. * p<0.05, ** p<0.01, **** p<0.0001. See also *Figure 3—figure supplement 1*.

The online version of this article includes the following figure supplement(s) for figure 3:

**Figure supplement 1.** Establishment of control and target windows, and representative example of all three activity parameters segmented by trial type and outcome.

*Figure 3—figure supplement 1A–D*). Thus, POm-striatal projections represent a learning-related signal that increases prior to and becomes stereotyped to texture presentation.

## Licking and pupil dilation also markedly increase across learning, but only pupil is correlated with calcium activity

Both licking and pupil dilation also exhibited marked increases across learning (*Figure 2G-K*, *Figure 2—figure supplement 1A-B*). Licks occurring before texture presentation decreased dramatically as licking became stereotyped within the PT window (*Figure 3D*). We assessed whether licking and POm activity were correlated as both increase across learning (*Figure 3A and D*) by plotting the average calcium activity from the Early, Late, and Expert time points with the grand average reaction time (RT) set to 0 (*Figure 3E*). Comparison of average calcium activity pre- and post-RT revealed an overall significant reduction (0.79±0.09 in pre-RT vs. 0.00±0.03 in post-RT, p=0.0221; *Figure 3F*), indicating no correlation between POm and licking activity, validating previous results (*La Terra et al., 2022*).

Pupil dilation started immediately following the cue sound and peaked near texture presentation before slowly decreasing throughout the rest of the trial (*Figure 3G*). Similar to previous studies (*Petty et al., 2021*; *La Terra et al., 2022*), we found that pupil and POm activity were tightly correlated but decoupled during the PT window with POm activity immediately regressing to baseline, while pupil activity remained elevated (*Figure 3A and G*). Pupil dilation lagged by ~250ms on average (*Figure 3H*), and the correlation became more consistent across learning and was restricted to before and at texture presentation (*Figure 3I*). Thus, despite all three activity parameters increasing across learning, POm activity is not related to motor activity and is only correlated with pre-PT pupil dilation.

## Increased POm axonal calcium activity is independent of trial type or outcome

This whole-session analysis did not account for sensory (texture) or outcome differences. To assess whether POm axonal activity is sensory-related (texture-specific), sessions were segmented by the presented texture (Go or NoGo; *Figure 3—figure supplement 1E–F*) and by trial outcome (below). Notably, licking and pupil dynamics have discrete activation patterns based on the presented texture (*Lee and Margolis, 2016*). Licking activity in the Go condition occurs squarely within the PT window and is indistinguishable from whole-session licking activity (*Figure 3D*), while licking peaks near the

end of the grace period in the NoGo condition (*Figure 3J*). Similarly, pupil activity starts to increase at the same point in both conditions but deviates at the end of the grace period. Conversely, POm axonal activity remained strikingly consistent between the Go and NoGo conditions (*Figure 3J*). This observation was further validated by comparing the average maximum calcium amplitude of each behavioral time point between the presented texture conditions. The main effect of behavioral time points was significant ($F_{(4,32)}$=57.16, p<0.0001), but the main effect of presented texture was not ($F_{(1,8)}$=0.3797, p=0.5549; *Figure 3K*). Post-hoc comparisons found that the maximum calcium amplitude during Shaping (Go: 0.55±0.06; NoGo: 0.53±0.04) was significantly reduced compared to Early Learning (Go: 1.18±0.11, p=0.0016; NoGo: 1.05±0.11, p=0.0128), Late Learning (Go: 1.65±0.23, p<0.0001; NoGo: 1.59±0.16, p<0.0001), and Expert sessions (Go: 1.68±0.21, p<0.0001; NoGo: 1.52±0.12, p<0.0001), but not the Reward session. The same overall effects were noted for the auROC analysis (main effect of behavioral time point: $F_{(4,32)}$=66.43, p<0.0001; main effect of presented texture: $F_{(1,8)}$=0.01694, p=0.8997; *Figure 3L*). As with the maximum amplitude, post-hoc comparison highlighted that the auROC value during Shaping (Go: 0.59±0.02; NoGo: 0.61±0.03) was significantly smaller than Early Learning (Go: 0.72±0.02, p=0.0035; NoGo: 0.70±0.04, p=0.0355), Late Learning (Go: 0.84±0.02, p<0.0001; NoGo: 0.83±0.02, p<0.0001), and Expert sessions (Go: 0.87±0.02, p<0.0001; NoGo: 0.85±0.02, p<0.0001), but not the Reward session. Thus, despite divergent licking and pupil activity as a function of the presented texture, axonal calcium activity remained unchanged, indicating POm-striatal projections do not encode vibrissae-specific sensory information.

We next tested whether POm calcium activity correlated with trial outcome (e.g. Hit, Miss, False Alarm, Correct Rejection; *Figure 2C–D*). Upon segmentation by trial outcome (*Figure 3—figure supplement 1G–J*), licking and pupil dynamics exhibited discrete outcome-dependent activity patterns (*Lee and Margolis, 2016*), but calcium activity again remained remarkably consistent (*Figure 3M–P*). These results were further validated by comparing the average maximum calcium amplitude of each behavioral time point between the outcome conditions: the main effects of behavioral time point ($F_{(1.273,5.090)}$=22.09, p=0.0043) and trial outcome ($F_{(1,153, 4.613)}$=10.80, p=0.0231; *Figure 3Q*) were significant. However, the only significant post-hoc comparisons were between Hit-Shaping and CR-Shaping (p=0.0305) and FA-Shaping and CR-Shaping (p=0.0114). Overall, POm-striatal projections do not appear to encode texture- or outcome-specific information, suggesting that POm may represent a behaviorally relevant arousal-related role.

## Optogenetic suppression of POm-striatal axons during behavior slows learning rate

Our findings indicate that POm-striatal inputs may represent an arousal signal. To investigate this possibility, wild-type mice were divided into two cohorts: a No Stim and a photoinactivation JAWS cohort. For both cohorts, a 200 μm fiber optic cannula was unilaterally implanted over left pDLS. However, only the JAWS cohort received an injection of the inhibitory actuator JAWS (*Figure 4A–B*; *Huda et al., 2020*; *Nguyen et al., 2021*; *Mo et al., 2023*) into ipsilateral POm, verified by the stereotypical POm-cortical projection pattern (*Figure 4B*).*Bureau et al., 2006*; *Meyer et al., 2010*; *Ohno et al., 2012* Both cohorts were trained on the Go/NoGo task. However, once the Learning phase started, POm-striatal axons of the JAWS cohort were photoinactivated for ~50% of trials per session until reaching the Expert phase. Photoinactivation occurred via an amber (617 nm) LED (~7 mW) for 2 s of constant illumination centered around the PT window start (*Figure 4C–D*) where the maximum calcium signals were detected (*Figure 3A*). Both the No Stim and JAWS cohorts achieved Expert status (*Figure 4—figure supplement 1A–B*) by increasing Hit Rate, decreasing FA Rate, and with licking becoming stereotyped across learning (*Figure 4E–H*, **S5A-B**). However, photoinactivation of POm-striatal axons during the Learning phase resulted in the JAWS cohort requiring significantly more sessions (12.50±0.50 sessions, n=4 mice) to attain Expert status compared to the other two cohorts with different experimental conditions (FP with constant illumination across the entire session: 7.60±0.51 sessions, n=5 mice; No Stim with no illumination: 7.40±0.24 sessions, n=5 mice) ($F_{(3,11)}$ = 8.542, p=0.0041, JAWS vs. FP p=0.0470; JAWS vs. No Stim p=0.0202; FP vs. No Stim p>0.9999; *Figure 4I–J*). We assessed whether photoinactivation modified behavioral responding parameters between the Learning and Expert phases (*Figure 3K*). The only significant effect was a decrease in overall responding (Bias) during the Learning phase (Off: 0.56±0.32, On: 0.48±0.32, p=0.0351). Due to this, we assessed if inhibition resulted in slower RTs and found that the average RT was significantly

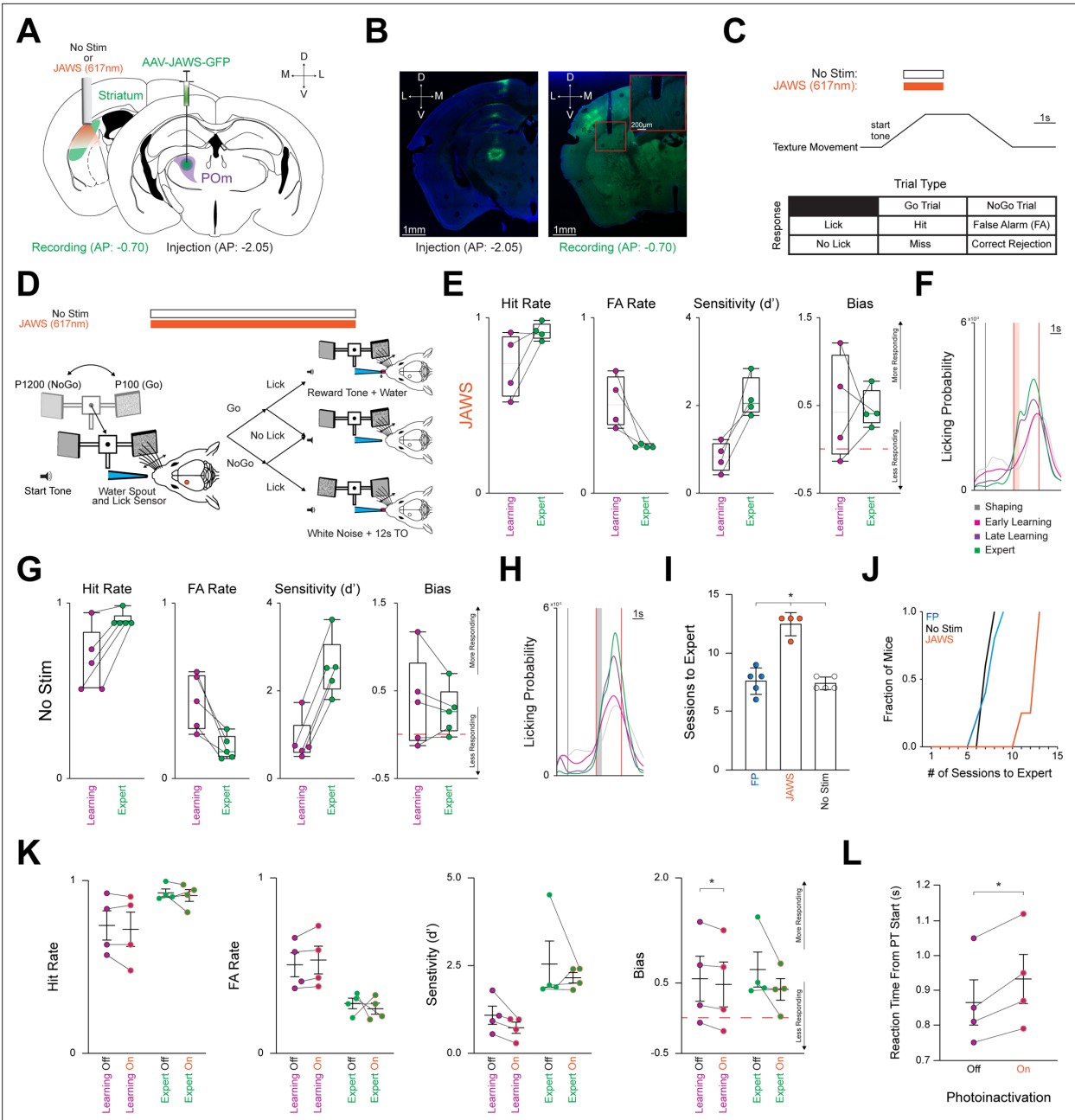

**Figure 4.** Photoinactivation increases number of sessions required For expert discrimination. (**A**) Schematic detailing pAAV-hSyn-JAWS-KGC-GFP-ER2 (JAWS) injection unilaterally into POm (*Right*) and a 200 μm cannula implanted in the left posterior DLS (*Left*). For the No Stim cohort, only the cannula was implanted in the left posterior DLS. Activation of the inhibitory JAWS opsin was performed constantly on for 1 s before and after texture arrival in the whisker field. JAWS activation probability per trial = 0.50. (**B**) Representative injection site in POm (*Left*), and the cannula placement in the posterior DLS along with ascending POm axons (*Right*). Scale = 1 mm. Red inset shows ascending POm axons underneath the optic cannula. Inset scale = 200 μm. (**C**) *Top:* Stimulation timing (constant illumination for 2 s, centered around texture arriving at its endpoint) and texture movement representation during a trial. Note that no light is presented for the No Stim cohort as no stimulation occurred. *Bottom:* Outcomes for each stimulus-response pair. (**D**) Schematic representing texture movement and potential outcomes during a single trial. (**E**) Changes in Hit Rate, FA Rate, Sensitivity (**d'**), and Bias of all JAWS cohort mice (n=4) as they transition from the Learning to the Expert phase in box-and-whisker plots. Note that mice are classified as Expert when they achieve a Hit Rate ≥0.80 and a FA Rate ≤0.30 for two consecutive sessions. Red line = 0. See *Figure 4—figure supplement 1*. (**F**) Probability density function for overall licking-related activity at each behavioral time point for the JAWS cohort. Vertical black line = sound cue representing trial start as the texture moves toward the whisker field. Vertical red lines = start (texture arrival at endpoint in whisker field) and end (texture departure toward starting point) of the PT window (time where mice can respond by licking). Colored boxes = 500ms grace period (licking does not trigger any outcomes). (**G–H**) Same as in (**E, F**) for the No Stim cohort. (**I**) JAWS cohort requires significantly more training sessions for expert discrimination

*Figure 4 continued on next page*

*Figure 4 continued*

compared to the FP and No Stim cohorts. (**J**) Longitudinal representation of sessions required for expert discrimination. (**K**) Comparison of Hit Rate, FA Rate, Sensitivity (**d'**), and Bias during the Learning and Expert phases. (**L**) Average RT is slower during photoinactivation than non-photoinactivated trials. Data are mean ± SEM. * p<0.05.

The online version of this article includes the following figure supplement(s) for figure 4:

**Figure supplement 1.** Individual longitudinal learning-related changes in behavioral parameters and licking activity for the JAWS and no stim cohorts, and whisker trim.

slower in the On condition (Off: 0.87±0.07, On: 0.93±0.07, p=0.0124). Thus, the suppression of this behaviorally relevant arousal signal resulted in more learning sessions due to delayed RTs.

## Discussion
### POm involvement in sensorimotor processing

POm has received considerable attention for its potential roles in sensory/nociceptive processing (*Moore et al., 2015*; *Diamond et al., 1992*; *Sosnik et al., 2001*; *Urbain et al., 2015*; *Chiaia et al., 1991a*; *Lavallée et al., 2005*; *Masri et al., 2008*; *Masri et al., 2006*), pain signaling (*Noseda et al., 2010*; *Frangeul et al., 2014*; *Osaki et al., 2022*; *Dallel et al., 1988*; *Gauriau and Bernard, 2004*), and cortical plasticity mediation (*Gambino et al., 2014*; *Audette et al., 2019*; *Williams and Holtmaat, 2019*; *Zhang and Bruno, 2019*; *La Terra et al., 2022*; *Gharaei et al., 2020*; *Audette et al., 2018*; *Qi et al., 2022*). Its widespread connectivity with sensory and motor cortical areas (S1, S2, and M1; *El-Boustani et al., 2020*; *Wimmer et al., 2010*; *Ohno et al., 2012*; *Viaene et al., 2011*; *Smith et al., 2012*; *Hooks et al., 2013*; *Luo et al., 2019*), along with its strong modulation with behavioral state and arousal (*Masri et al., 2008*; *Petty et al., 2021*), suggests a significant, yet still undetermined role in sensory-guided behaviors. VPM thalamus, by contrast, projects to much more delimited domains of S1, with axons coursing through striatum without forming synapses (*Hunnicutt et al., 2016*). POm projections to the dorsal striatum arise from branches of the main axons bound for the cortex and terminate in a region of the dorsal striatum that overlaps with corticostriatal inputs from S1 and M1 (*El-Boustani et al., 2020*; *Alloway et al., 2003*; *Groh et al., 2014*; *Yamawaki and Shepherd, 2015*; *Sumser et al., 2017*; *Ohno et al., 2012*; *Hunnicutt et al., 2016*; *Casas-Torremocha et al., 2022*), potentially placing POm projections in a powerful position to influence sensorimotor integration. We aimed to elucidate the functional innervation pattern of POm onto striatal cell types, and the role of POm-striatal projections on behavioral performance and learning. Slice electrophysiology revealed strong POm-mediated synaptic inputs to D1-SPNs, D2-SPNs, and PV interneurons, with shorter latency responses in PVs. In vivo photometry recordings showed increasing activation of POm-striatal axons across task learning, with axonal signals and pupil dilation in expert mice increasing prior to and becoming stereotyped to stimulus presentation, independent of stimulus type or trial outcome. Photoinactivation of POm-striatal axons resulted in prolonged RTs and delayed learning, with more training sessions required to achieve expert behavioral performance. We discuss below the complex feedforward and feedback circuitry that POm is part of, and potential functional differences in POm-striatal projections compared to POm-cortical projections in sensorimotor behavior (*Petty et al., 2021*; *La Terra et al., 2022*; *Qi et al., 2022*). We propose that POm could provide an early salience- or arousal-related 'priming' signal to the pDLS that interacts with subsequent corticostriatal signals, leading up to behavioral action selection.

### Does POm provide a 'priming' signal to striatum?

A striking feature of our POm-striatal projection measures in behaving mice was the early task-related activation, with the strongest increases before and during texture presentation rather than during trial outcome, reward presentation, or licking. SPNs require an up-state transition from their relatively hyperpolarized resting potentials to reach action potential threshold (*Wilson, 1993*). While this up-state transition is correlated with cortical oscillatory activity (*Ding et al., 2008*; *Stern et al., 1997*), the role of thalamostriatal projections has not been fully elucidated (*Reig and Silberberg, 2014*). POm inputs likely arrive in striatum with shorter latency than cortical inputs following whisker stimulation (*Smith et al., 2012*; *Alloway et al., 2017*; *Alloway et al., 2014*; *Mowery et al., 2011*), and

therefore could be involved in initiating the up-state transition in the SPNs they innervate. We refer to this early POm-striatal input as a putative 'priming' signal because it could serve to sensitize striatal neurons to respond to subsequent synaptic input. We found that POm provides large amplitude PSPs to D1- and D2-SPNs, of similar amplitude to previously measured M1 inputs and larger amplitude than S1 inputs (*Lee et al., 2019*). This equally strong innervation of D1- and D2-SPN is consistent with previous work on the co-activation of SPN populations during natural movements (*Cui et al., 2013*; *Tecuapetla et al., 2014*; *Klaus et al., 2017*). The early timing, together with the equal large-amplitude innervation of both direct and indirect pathways (D1- and D2-SPNs and PV interneurons), suggest that POm projections are well-positioned to 'prime' dorsal striatal circuitry for processing subsequent synaptic input.

## Impact of PV-interneuron-mediated feedforward GABAergic signaling

POm innervation of PV interneurons was particularly prominent, providing equal amplitude PSPs with shorter latency than SPNs (*Figure 1*). Striatal PV cells are thought to provide feedforward inhibition to surrounding neurons via short-latency GABAergic synaptic signals (*Tepper et al., 2018*) How could PV cells be involved in 'priming'? First, PV cells do not exert an inhibitory effect, as defined by postsynaptic hyperpolarization, under all conditions. Stimulation of a presynaptic PV cell in synaptically coupled PV-SPN paired recordings causes GABA-mediated depolarization in the postsynaptic SPN when the SPN's resting potentials was negative (*Koós and Tepper, 1999*). Indeed, recent work found that the chloride equilibrium potential is surprisingly positive in adult striatal SPNs, closer to –60 mV compared to the more typical –80 mV (*Day et al., 2024*) An approximately –60 mV equilibrium potential means that a GABA receptor-mediated chloride current produces depolarization instead of hyperpolarization when the postsynaptic neuron is at a down-state resting potential of approximately –80 mV (*Day et al., 2024*; *Koós and Tepper, 1999*; *Fino et al., 2018*). In *Day et al., 2024*, when GABA-mediated depolarization was paired with further glutamate-evoked depolarization, the probability of action potential firing was increased, suggesting that GABA-mediated depolarization was excitatory rather than shunting (*Day et al., 2024*). In this scenario, if SPNs are sitting at a relatively hyperpolarized (down state) resting potential when POm becomes active, the disynaptic GABAergic input provided by PV interneurons could actually have a depolarizing effect on SPNs and combine with the direct glutamate-mediated depolarization provided by POm. Thus, under certain conditions, PV interneurons could contribute to 'priming' striatal circuitry by adding to the early depolarization of SPNs.

The situation is different when the postsynaptic neuron is already depolarized above –60 mV (as in an up state). In this scenario, the effect of GABAergic input is hyperpolarizing, and PV-cell-mediated feedforward signals to SPNs would be inhibitory, in line with traditional views (*Gittis et al., 2010*; *Lee et al., 2017*; *Owen et al., 2018*). In this case, instead of enhancing the initial depolarization of striatal neurons (as above), POm input would likely provide a temporally restricted depolarizing signal, followed by hyperpolarization. POm-mediated feedforward inhibition could, depending on the intricacies of local connectivity, act to prevent up-state transitions of specific SPN ensembles, effectively increasing the signal-to-noise of striatal population activity and therefore the recruitment of SPNs by subsequent cortical or thalamic inputs. Thus, it is possible that POm projections have dual effects on striatal circuits depending on ongoing striatal activity. GABAergic signaling, mediated by PV or other types of striatal interneurons, is likely critical in mediating such state-dependent effects on postsynaptic striatal circuitry.

## POm involvement in salience networks and behavioral state

Many extrinsic and intrinsic factors influence striatal function. For example, the striatum receives inputs from myriad subcortical regions including other thalamic nuclei (*Hunnicutt et al., 2016*; *Mandelbaum et al., 2019*) and external globus pallidus (*Mastro et al., 2014*; *Baker et al., 2023*). A crucial factor in POm signal timing is its inhibitory gating by two GABAergic inputs, ventral zona incerta (vZI; *Lavallée et al., 2005*; *Roger and Cadusseau, 1985*; *Power et al., 1999*; *Barthó et al., 2002*; *Watson et al., 2015*) and anterior pretectal (APT; *Lavallée et al., 2005*; *Giber et al., 2008*; *Bokor et al., 2005*). All three nuclei receive ascending spinal trigeminal (whisker-related sensory) inputs, but vZI and APT efficiently shunt incoming sensory information via feedforward inhibition onto POm neurons (*Lavallée et al., 2005*; *Trageser and Keller, 2004*; *Trageser et al., 2006*). This inhibitory gating is overcome

by (1) arousal-related cholinergic suppression of presynaptic GABA release within POm (*Masri et al., 2006*) or (2) convergence of bottom-up and top-down signals within a specified temporal window (*Groh et al., 2014*; *Miller-Hansen and Sherman, 2022*). Both factors are likely at play during sensory-guided behavior. The involvement of cholinergic signaling is consistent with the POm-striatal 'priming' concept if cholinergic signaling prompts POm before a behaviorally relevant period of sensation, or immediately prior to sensory information becoming perceptible (*Sachidhanandam et al., 2013*). In addition to cholinergic, cortical, and subcortical GABAergic afferents, POm also receives direct glutamatergic input from superior colliculus (SC) that enhances sensory responses (*Roger and Cadusseau, 1984*; *Gharaei et al., 2020*). SC, a region implicated in attentional orienting of either somatosensory (*Gharaei et al., 2020*; *Hemelt and Keller, 2008*) or visually relevant (*Schneider, 1969*; *Ahmadlou et al., 2018*) stimuli, bidirectionally modulates POm, further ascribing a potential arousal-related functional role. It is yet to be resolved to what extent POm-striatal inputs are driven by ascending (feedforward) sensory information, descending cortical-POm feedback, or the convergence of both, but the interplay and integration of ascending and descending signals is likely to be an essential feature in POm activation and its involvement in behavioral salience.

It is difficult to differentiate movement- and arousal-related brain states. POm neuronal activity shows a close relationship with spontaneous (task-free) whisker movements, and pupil-indexed arousal in head-restrained mice (*Petty et al., 2021*) while in freely moving mice both VPM and POm activity correlate with head and whisker movements, (*Oram et al., 2024*) indicating that POm is generally coactive with exploratory head and whisker movements and behavioral arousal. During task performance, the situation may change with training and attentional effects. For example, *Petty and Bruno, 2024* showed that POm activity correlates more closely with task demands than tactile or visual stimulus modality. Our data indicate that POm-striatal axonal signals are increased at trial start during anticipation of tactile stimulus delivery and through the sensory discrimination period, then decrease to baseline levels during licking and water reward collection (*Figure 3*). Results of *Petty and Bruno, 2024* together with ours suggest that POm is particularly active during the context of behaviorally relevant task performance. Thus, we think that pupil dilation indexes general movement and arousal, which POm correlates with during spontaneous behaviors, but that POm activity becomes more specific and more prominent during the movement, anticipation, and arousal associated with learned sensorimotor behavioral tasks.

## Parallel thalamostriatal pathways, additional striatal cell types

Pf is the best studied thalamic nucleus projecting to the striatum. A major difference between POm and Pf striatal innervation is Pf innervation of cholinergic (ChAT; tonically active) interneurons (*Guo et al., 2015*; *Ding et al., 2008*; *Johansson and Silberberg, 2020*; *Brown et al., 2010*; *Díaz-Hernández et al., 2018*; *Klug et al., 2018*; *Poppi et al., 2021*). We did not determine whether POm innervates ChAT interneurons, but available evidence suggests it might only weakly. Retrograde tracing to map brain-wide inputs to striatal ChAT interneurons showed that Pf provides the majority of thalamic innervation of striatal ChAT neurons compared to other thalamic nuclei (*Klug et al., 2018*). Many other thalamic nuclei, including POm, showed little or no labeling. However, it is possible that these thalamic nuclei, including POm, provide functional innervation of ChAT interneurons that is insufficiently assessed by anatomical methods alone.

Another difference between Pf and other thalamic nuclei innervation of striatum is the subcellular localization of their synaptic inputs. Pf synapses tend to localize to dendritic shafts of SPNs rather than dendritic spines (*Dubé et al., 1988*; *Wright et al., 1999*; *Raju et al., 2008*), whereas other thalamic nuclei form synapses onto SPN dendritic spines (*Smith et al., 2014*). These differences could affect postsynaptic signal processing, such that Pf shaft inputs may induce a longer temporal window for integration via dendritic filtering (*Häusser, 2001*; *Day et al., 2008*), whereas POm spine inputs may cause a more restricted temporal window for signal integration. These properties could influence the behaviorally relevant time window of thalamostriatal and corticostriatal signal integration (*Sanabria et al., 2024*; *Lee et al., 2019*; *Johansson and Silberberg, 2020*), for example, with POm requiring time-locked inputs onto the same or nearby spines for maximal postsynaptic effect (*Alloway et al., 2017*). The occurrence and relevance of such mechanisms in different thalamostriatal pathways will require further investigation. Additionally, it will be important to understand the innervation patterns of POm-striatal projections beyond the three cell types we have studied here,

including other interneurons responsive to thalamic stimulation such as tyrosine-hydroxylase (THIN) and neurogliaform (NGF) interneurons (*Tepper et al., 2018*; *Assous and Tepper, 2019*). Lastly, the effects of dopamine modulation of thalamostriatal signaling directly (*Tritsch and Sabatini, 2012*) and indirectly via cholinergic interneurons (*Reynolds et al., 2022*; *Matityahu et al., 2023*) would be an important area of further study.

## POm as a link between cortical and striatal signaling

POm axons bifurcate and contact neurons within pDLS (*Ohno et al., 2012*; *Spreafico et al., 1987*; *Descheˆnes et al., 1995*; *Viaene et al., 2011*; *Smith et al., 2012*; *Li et al., 2020*; *O'Reilly et al., 2021*), but the main terminations continue to cortex where they strongly innervate L1 and L5a of S1 (*Wimmer et al., 2010*; *Ohno et al., 2012*; *Viaene et al., 2011*; *Smith et al., 2012*). POm input excites L5 pyramidal neurons (and L2/3 neurons) via L5 perisomatic synapses as well as distal dendritic synapses in L1 (*Audette et al., 2018*). Synaptic plasticity of POm-cortical inputs and increased excitation of L2/3 and L5 neurons has been implicated in sensory learning (*Gambino et al., 2014*; *Audette et al., 2019*). Given the capacity of POm-cortical synapses for learning-related plasticity, it will be important to determine whether POm-striatal synapses also undergo learning-related plasticity and how it may be distinct from POm-cortical plasticity. Remarkably, POm axonal branches, even from the same neuron, display synaptic terminals with distinct structure and functional properties in M1 compared to S1 (*Rodriguez-Moreno et al., 2020*); thus it is possible that the striatal synapses of POm neurons could undergo distinct changes with learning compared to their cortical counterparts.

How the divergent cortical and thalamic projections of POm neurons may be involved in coordinating thalamostriatal and corticostriatal signals is another important issue. Striatal neurons that receive POm input are likely also to receive cortical L5a input, because L5a neurons comprise the predominant S1-striatal projection (*Pan et al., 2010*; *Wall et al., 2013*; *Hooks et al., 2018*). Thus a disynaptic POm-S1-pDLS loop exists in addition to the monosynaptic POm-pDLS projection. During behavior, striatal circuitry may be initially engaged via the monosynaptic POm projection followed, in tens of milliseconds, by the disynaptic loop that recruits L5a cortical pyramidal neurons to provide input to already depolarized ('primed') striatal neurons. On the hundreds-of-milliseconds time scale of a behavioral trial, it is possible that these pathways are active multiple times, with L5b cortical feedback activating POm (*Groh et al., 2014*; *Sumser et al., 2017*) after the initial feedforward volley of activity, initiating POm-driven signaling again. It is possible that an individual SPN receives both POm and L5a input, with the POm input providing the initial 'priming' depolarization, followed by further depolarizing input from a POm-driven L5a neuron. It is unclear whether thalamic 'priming' of striatum is necessary for corticostriatal activation of SPNs. Based on the recurrence and convergence of these pathways, it is possible that POm acts to synchronize thalamo- and corticostriatal signals under certain behaviorally relevant conditions. This would be a unique role for POm compared to Pf, since Pf strongly innervates striatum but only weakly innervates cortex.

## POm-striatal recruitment during learning: unresolved mechanisms

Behavioral state (*Petty et al., 2021*) influences the efficiency of sensorimotor learning (*La Terra et al., 2022*; *Qi et al., 2022*). Overall, we observed a learning-related increase in POm-striatal axonal activity (as measured by photometry) that correlated with pupil dilation in many but not all phases of behavioral performance, and a necessity of these projections for efficient behavioral performance and learning, supporting a role in task-related behavioral arousal. Photometry signals and pupil dilation in expert mice were tightly correlated between trial onset and texture presentation, but decoupled when photometry signals decreased prior to licking and reward delivery. Sorting trials by presented texture (stimulus) or trial outcome (response), photometry signals remained elevated, while both licking and pupil dilation exhibited stimulus- or response-specific changes. This suggests that POm-striatal projections do not encode specific sensory- or outcome-related information, but rather arousal or salience during anticipatory states of a learned behavior. The effects of photoinactivation of PO-striatal projections included delayed RTs on individual trials and more sessions required to achieve expert performance. These effects suggest that POm serves to enhance striatal signaling during increased behavioral states associated with performance of learned sensorimotor behavior. POm activation early during behavioral trials may set the stage for other important inputs, such as corticostriatal inputs, to influence action selection. The increased amplitude of POm-striatal signals with learning suggests an

increased influence of POm on striatum, and thereby on goal-driven sensorimotor behaviors. POm likely becomes increasingly tuned to task-specific behavioral arousal and anticipation than to general movement-related arousal.

An important unresolved issue is how POm-striatal signals increase with learning. Inherent in our measures of POm-striatal axons is the limitation that photometry only captures a bulk axonal signal and lacks the ability to resolve individual axons. Therefore, we were unable to resolve whether the increased bulk signal that we observed was due to a uniform increase across all POm-striatal axons or a heterogeneous increase present only in specific axons. This is important because POm neurons exhibit heterogeneous responses to direct paralemniscal stimulation (*Mo et al., 2017*), peripheral stimulation (*Chiaia et al., 1991a*) and the suppression of vZI activity (*Trageser and Keller, 2004*) and also show functionally distinct anterior and posterior subpopulations (*El-Boustani et al., 2020*; *Ohno et al., 2012*; *Viaene et al., 2011*; *Casas-Torremocha et al., 2022*; *Arai et al., 1994*). Therefore, we might expect heterogeneous learning-related changes in activity within the POm neuronal population, a mechanistic question that would be important to address in future studies.

Although we placed the optical fiber above pDLS to specifically record from axons, rather than somas, it is possible that POm-cortical axons of passage contributed to the recorded signal. However, even if signal contamination were present, we think the measures would be largely similar, as most POm-cortical projections bifurcate within striatum rather than projecting exclusively to either striatum or cortex (*Ohno et al., 2012*; *Smith et al., 2012*). Signal modification is more likely to occur at synapses via presynaptic or postsynaptic mechanisms than at the main axon (*Debanne, 2004*; *Citri and Malenka, 2008*; *Kreitzer and Malenka, 2008*). In future work, it would be valuable to investigate the intriguing possibility that POm projections have discrete target-specific functions, such that POm-striatal signals may be involved in distinct aspects of sensorimotor behavior compared with POm-cortical signals (*Petty et al., 2021*; *La Terra et al., 2022*; *Qi et al., 2022*).

In summary, we show that POm-striatal projections encode a behaviorally relevant arousal-related signal that increases with learning. POm-striatal projections may constitute a 'priming' signal that combines with corticostriatal signals and contributes to inducing the up-state transition of SPN ensembles involved in action selection, by equally engaging D1- and D2-SPNs and PV interneurons. These finding suggest a previously unknown functional role of POm engagement of striatal microcircuitry. It will be important for future studies to investigate whether POm further innervates other striatal interneurons, to assess the timing between POm-striatal and corticostriatal inputs to SPNs, and to assess whether POm-striatal synapses undergo synaptic plasticity across learning.

# Materials and methods

**Key resources table**

| Reagent type (species) or resource | Designation | Source or reference | Identifiers | Additional information |
|---|---|---|---|---|
| Transfected construct (*Chlamydomonas reinhardtii*) | pAAV-hSyn-hChR2(H134R)-EYFP | Addgene | RRID:Addgene_26973 | Viral vector to express channelrhodopsin-2 |
| Transfected construct (synthetic) | pAAV-hSynapsin-axon-jGCaMP8s-P2A-mRuby3 | Addgene | RRID:Addgene_172921 | Viral vector to express GCaMP8s |
| Transfected construct (Haloarcula salinarum) | pAAV-hSyn-JAWS-KGC-GFP-ER2 | Addgene | RRID:Addgene_65014 | Viral vector to express JAWS |
| Strain, strain background (*Mus musculus*, C57/Bl6, male and female) | Mouse: B6.FVB(Cg)- Tg(DrD1creEY262Gsat/Mmucd) | Gensat/MMRRC | RRID:MMRRC_030989_UCD | |
| Strain, strain background (*Mus musculus*, C57/Bl6, male and female) | Mouse: B6.FVB(Cg)-Tg(Adora2a-cre)KG139Gsat/Mmucd | Gensat/MMRRC | RRID:MMRRC_036158_UCD | |
| Strain, strain background (*Mus musculus*, C57/Bl6, male and female) | Mouse: B6.129P2-Pvalb[tm1(cre)Arbr]/J | Jackson Laboratory | RRID:IMSR_JAX:017320 | |

*Continued on next page*

*Continued*

| Reagent type (species) or resource | Designation | Source or reference | Identifiers | Additional information |
|---|---|---|---|---|
| Strain, strain background (*Mus musculus*, C57/Bl6, male and female) | Mouse: B6.Cg-Gt(ROSA)26Sor$^{tm14(CAG-tdTomato)Hze}$/J | Jackson Laboratory | RRID:IMSR_JAX:007914 | |
| Strain, strain background (*Mus musculus*, C57/Bl6, male and female) | Mouse: Wild type | Jackson Laboratory | RRID:IMSR_JAX:000664 | |
| Software, algorithm | Patchmaster Next | HEKA | http://www.heka.com/ | |
| Software, algorithm | Adobe Illustrator | Adobe | https://www.adobe.com/ | |
| Software, algorithm | Prism | GraphPad | https://www.graphpad.com/ | |
| Software, algorithm | Synapse | Tucker-Davis Technologies | https://www.tdt.com/ | |
| Software, algorithm | Spyder | Spyder IDE | https://www.spyder-ide.org/ | |
| Software, algorithm | LabVIEW | National Instruments | https://www.ni.com/en.html | |
| Software and algorithms | MATLAB | MathWorks | https://www.mathworks.com/ | |
| Software, algorithm | DeepLabCut | *Mathis et al., 2018* | N/A | |
| Software, algorithm | Scikit-image | *van der Walt et al., 2014* | N/A | |
| Software, algorithm | Electrophysiology Feature Extraction Library (eFEL) | *Ranjan et al., 2023* | N/A | |
| Software, algorithm | Zen Imaging Suite | Zeiss | https://www.zeiss.com/corporate/en/home.html | |
| Software, algorithm | Analysis scripts | Github | https://github.com/margolislab/Yonk_2024/ | |
| Other | Electrophysiology Vibratome | Leica | VT-1200S | Equipment |
| Other | Electrophysiology Pipette Puller | Sutter Instruments | P-1000 | Equipment |
| Other | Electrophysiology Amplifier | HEKA | EPC10USB | Equipment |
| Other | Fixed Upright Microscope | Olympus | BX51WI | Equipment |
| Other | Fiber Photometry Amplifier | Tucker-Davis Technologies | RZ10X Lock-In | Equipment |
| Other | Micropipette Puller | Sutter Instruments | P-30 Vertical Puller | Equipment |
| Other | Confocal Microscope | Zeiss | LSM800 Airyscan | Equipment |
| Other | Histology Vibratome | Leica | VT-1000 | Equipment |

## Animals

All work involving animals including housing, surgery, behavioral experimentation, and euthanasia was approved by the Rutgers Institutional Animal Care and Use Committee (protocol #: 999900197). Mice were group housed in a reverse light cycle room (lights on from 20:00 to 08:00) with food and water available ad libitum with the exception of mice undergoing water restriction during behavioral experiments. All handling and experiments were conducted within the dark phase of this light cycle. Regardless of their experimental designation, all experimental animals underwent a unilateral AAV injection or a simultaneous AAV injection and cannula implant between –5 and 65 days (average: 52.37 days, range: –7–61 days). Briefly, male and female double transgenic mice were used for electrophysiology experiments. To identify specific neuronal populations during electrophysiology,

Ai14 mice (cre-dependent tdTomato; The Jackson Laboratory, #007914) were crossed with either (1) D1-SPN cre (B6.FVB(Cg)-Tg(DrD1cre)EY262Gsat/Mmucd; MMRRC, #030989), (2) D2-SPN cre (B6.FVB(Cg)-Tg(Adora2a-cre)KG139Gsat/Mmucd; MMRRC, #036158), or (3) PV-cre (B6.129P2-Pvalb$^{tm1(cre)Arbr}$/J; The Jackson Laboratory, #017320) mice. This permitted red fluorescence in the specific cell population for visual identification via epifluorescent illumination. Animals designated for electrophysiology were euthanized between –2 and 4.5 months. Both male and female wild-type mice (The Jackson Laboratory #000664) were used for fiber photometry and optogenetic experiments. To motivate behavioral performance, daily water intake was restricted to ~1.5 mL per mouse per day. Body weight was carefully controlled and never permitted to drop below 80% of a calculated baseline value (*Guo et al., 2014*). The number of male and female mice were as follows, by experiment type: six male, four female (electrophysiology); three male, two female (fiber photometry); four male, five female (optogenetics). Data were not analyzed for sex differences.

## Electrophysiology and adeno-associated viral (AAV) injection

Male and female double transgenic mice designated for electrophysiology experiments underwent a unilateral injection of channelrhodopsin-2 (ChR2; pAAV-hSyn-hChR2(H134R)-EYFP; Addgene #26973; *Lee et al., 2010*) targeting the left POm thalamic nucleus. Briefly, mice were anesthetized using isoflurane (4% induction, 1–2% maintenance) and moved into a stereotaxic apparatus (Stoelting/Kopf Instruments) containing a feedback-controlled heating pad on the base (maintained between 35°C and 37°C; FHC). Ophthalmic ointment (Akorn) was applied to the eyes to prevent them from drying out. Ethiqa XR (3.25 mg/kg; Fidelis Animal Health) and Bupivacaine (0.25%, 0.1 mL, Fresenius Kabi) were injected subcutaneously into the right flank and scalp, respectively. After, the scalp was sterilized by three cycles of Betadine (Purdue Products) and 70% ethanol, a midline incision was made. The exposed skull was cleared of fascia and leveled by confirming that bregma and lambda coordinates were on the same dorsoventral plane. A craniotomy was drilled (coordinates with respect to bregma: anteroposterior (AP)=–2.05 mm, mediolateral (ML) = +1.35 mm, dorsoventral (DV)=–3.25 mm) and the micropipette containing ChR2 was slowly lowered to the appropriate depth and permitted to sit for 5 min. Following this, ~100 nL of ChR2 viral solution was injected over the course of ~15 min via the Nanoject III system (Drummond Scientific). After an additional delay of 12 min, the micropipette was slowly raised, the scalp was sutured and secured with tissue glue. Immediately following surgery, mice were transferred to a clean cage on top of a heating blanket until ambulation was observed. Mice were continually monitored for 72 hr post-surgery. After this observation period, mice were permitted to recover for at least 3 weeks before undergoing electrophysiological experiments, permitting ChR2 expression into POm axon terminals in the striatum.

## Whole cell patch clamp recordings

Mice were briefly induced (via 3% isoflurane), deeply anesthetized with an intraperitoneal injection of ketamine-xylazine (300/30 mg/kg, respectively), and transcardially perfused with recovery artificial cerebrospinal fluid (ACSF), which contains the following (in mM): NMDG 103, KCl 2.5, NaH$_2$PO$_4$ 1.2, NaHCO$_3$ 30, HEPES 20, Glucose 25, HCl (1 N) 101, MgSO$_4$ 10, Thiourea 2, Sodium Pyruvate 3, N-Acetyl-L-Cysteine 12, and CaCl$_2$ 0.5 (saturated with 95% O$_2$ and 5% CO$_2$; *Lee et al., 2019*; *Assous and Tepper, 2019*). Following decapitation, the brain was rapidly extracted and submerged in recovery ACSF before being mounted onto a VT-1200S vibratome (Leica). The vibratome chamber was filled with oxygenated recovery ACSF, and 300 μm slices were cut. Slices were immediately transferred to recovery ACSF that was heated to 35 °C for ~5 min. After, slices were transferred to RT external ACSF which contained (in mM): NaCl 124, KCl 2.5, NaHCO$_3$ 26, NaH$_2$PO$_4$ 1.2, Glucose 10, Sodium Pyruvate 3, MgCl$_2$ 1, and CaCl$_2$ 2 (saturated with 95% O$_2$ and 5% CO$_2$), and slices were allowed to recover for at least ~45 min before recording. Once the hippocampus began to appear, vibratome sectioning was terminated, and the posterior tissue block containing the injection site was transferred into 10% neutral-buffered formalin for post-hoc confirmation.

Whole-cell patch clamp recordings were acquired from slices that were constantly superfused (2–4 mL/min) with oxygenated external ACSF at ~34 °C. Slices and cells were visualized by infrared differential interference contrast (IR-DIC) microscopy using a CCD camera (Hamamatsu) mounted onto a BX-51WI upright microscope (Olympus) fitted with a swinging objective holder containing two switchable lenses: a 4 X air lens and a 40 X water-immersion lens. Patch pipettes (3–5 MΩ) were pulled

from borosilicate glass micropipettes (2 mm O.D., Warner Instruments) using a P-1000 horizontal puller (Sutter Instruments).

Current-clamp recordings were obtained from unidentified and identified striatal neurons in mice expressing tdTomato in either D1-SPNs, D2-SPNs, or PV cells within pDLS (−0.34 to –1.22 mm relative to bregma), which is known to receive POm projections (*Smith et al., 2012*; *Alloway et al., 2017*; *Alloway et al., 2014*). The internal pipette solution for current-clamp experiments contained (in mM): K Methanesulfonate 130, KCl 10, HEPES 10, $MgCl_2$ 2, $Na_2ATP$ 4, $Na_2GTP$ 0.4, at pH 7.25 and 285–290 mOsm/L. Further, 2% biocytin was freshly dissolved in the internal solution on the recording day. ChR2 in the POm axon terminals was stimulated via illumination with a 2.5ms, 470 nm LED light pulse (~0.6 mW measured after the objective; Thorlabs) delivered through the 40 X objective lens. The illumination spot size had a diameter of ~550 μm. After patching, the internal solution was permitted to dialyze for ~5 min. At this point, the resting membrane potential was recorded. All cells were held at –80±2 mV to ensure equal driving forces when studying synaptic strength and short-term synaptic plasticity. Baseline voltages that drifted outside of this range were excluded from analysis. Patched cells were held for ~25–35 min while being run through a standardized set of protocols: (1) hyper-polarizing/depolarizing current steps, (2) single pulse (SP), (3) paired pulse ratio (PPR), and (4) train stimulation. Occasionally, unidentified cells within the same FOV were sequentially patched following the initial identified cell patch to control for injection site volume and location. After breaking in, the cell was allowed to recover for ~5 min before being subjected to hyperpolarizing and depolarizing current steps (–300 pA to +400 pA, 50 pA steps, 500ms, 15 sweeps) for cell health and intrinsic parameter confirmation. For the SP protocol, a single 2.5ms blue light pulse was presented once every 15 s for 20 sweeps. For the PPR protocol, five 2.5ms pulses, separated by 50ms inter-pulse intervals (IPI), were presented once every 15 s for 20 sweeps. For the train protocol, thirty 2.5ms pulses, separated by 64.2ms IPI (15 Hz), were presented once every 30 s for five sweeps. While these protocols were being run, biocytin within the internal solution diffused into the cell for subsequent 3D morphological reconstructions. Data were acquired via a EPC10USB amplifier and digitized at 20 kHz in Patchmaster Next (HEKA). Liquid junction potential was not corrected in these traces.

## Analysis of patch clamp recordings

All data were analyzed offline using custom-written MATLAB and Python scripts. DAT files from Patchmaster Next (HEKA) were imported, organized, and saved as a mat variable. The mat variable data was imported into Python for post-processing using the electrophysiology feature extraction librar-IFEL created by the Blue Brain Project (*Ranjan et al., 2023*). All pertinent intrinsic value parameters were calculated at the +350 pA current step. Key values pertaining to every action potential in a sweep were calculatlia eFEL including the action potential threshold value and index, the peak value and index, and the corresponding minimum afterhyperpolarization (AHP) value and index. Each action potential threshold value and index were captured using a derivative threshold method (dV/dt ≥10 mV/s). Action potential properties were assessed for all spikes in a sweep and averaged together. The action potential peak was defined as the difference between the action potential threshold and its maximum positive peak. Half-height width (HHW) and rise time were both calculated via interpolation. HHW was measured at 50% of the action potential peak, while rise time was measured from 10% to 90% of the action potential peak. AHP amplitude was calculated as the difference between the action potential threshold value and the minimum AHP value. Maximum frequency.

For all optogenetically evoked PSPs, a baseline measure was calculated by averaging the first 10,000 sampling points for each individual sweep. This measured value was subtracted from every value in each individual sweep, setting the baseline equal to zero. The maximum PSP amplitude, relative to the zero baseline value, was calculated from the average of 20 sweeps during SP and PPR protocols, and 5 sweeps during Train protocols. The latency to maximum PSP amplitude was measured as the difference between photostimulation onset and the maximum PSP index. For PPR and Train protocols, an exponential function was fitted to the decay component of each PSP, and their amplitudes were extracted after subtracting the decay of preceding PSPs.

## Simultaneous AAV injection and optical cannula implantation surgery

For male and female wild-type in the designated fiber photometry cohort, a 400 μm core optical cannula (ferrule OD = 2.5 mm, length = 2 mm, nA = 0.50; Thorlabs #CFM15L02) was implanted

directly above the left pDLS along with a~100 nL unilateral injection of axon-jGCaMP8s (pAAV-hSynapsin-axon-jGCaMP8s-P2A-mRuby3; Addgene #172921; *Broussard et al., 2018*) into the ipsilateral POm. Male and female wild-type mice designated for optogenetic manipulation were split into two cohorts: (1) the No Stim or (2) the JAWS cohort. Both optogenetic cohorts were implanted with a 200 μm core optical cannula (ferrule OD = 1.25 mm, length = 2 mm, nA = 0.50, CFMLC52L02, Thorlabs) above the left pDLS along with a~100 nL AAV injection into the ipsilateral POm. However, the No Stim cohort was injected with the excitatory optogenetic actuator, ChR2 (pAAV-hSyn-hChR2(H134R)-EYFP, Addgene #26973; *Lee et al., 2010*) while the JAWS cohort was injected with the inhibitory optogenetic actuator, JAWS (pAAV-hSyn-JAWS-KGC-GFP-ER2, Addgene #65014; *Huda et al., 2020*; *Nguyen et al., 2021*; *Mo et al., 2023*; *Chuong et al., 2014*). Further, an HHMI head plate (*Huber et al., 2012*) was fitted to each mouse using methods previously described (*Lee et al., 2019*; *Lee and Margolis, 2016*; *Margolis et al., 2012*; *Chen et al., 2013b*; *Chen et al., 2015*). Briefly, mice were anesthetized with isoflurane (4% induction, 0.8–1.5% maintenance) and mounted within a stereotaxic frame (Stoelting/Kopf Instruments) with a feedback-controlled heating pad (FHC) maintaining the body temperature between 35–37°C. Following administration of an analgesic (Ethiqa XR, 3.25 mg/kg; Fidelis Animal Health) to the left flank and a local anesthetic (0.25% Bupivacaine; Fresenius Kabi) under the scalp, the scalp was sterilized with a triple cycle of Betadine (Purdue Products) followed by 70% ethanol (Fisher). A midline incision was made, and a circular piece of scalp was removed to expose the skull. Both lateral muscles and the nuchal muscle were separated from the skull. The skull was cleaned by gently scraping away the periosteum. A dental etch bonding agent (iBond; Heraeus Kulzer) was applied to the clean skull and cured with blue light for 20 s twice. A ring of dental composite (Charisma; Heraeus Kulzer) was applied to the outer edge of the skull and cured with blue light. After ensuring that bregma and lambda coordinates were equal in the dorsoventral plane, two craniotomies were made: one above the left posterior striatum (AP = –0.80 mm, ML = +2.8 mm, DV = 1.8 mm) for cannula implantation and the other above left POm (AP = –2.05 mm, ML = +1.35 mm, DV = –3.25 mm) for the corresponding AAV injection. The AAV injection was always performed first. The micropipette containing axon-jGCaMP8s (fiber photometry cohort), ChR2 (No Stim cohort), or JAWS (JAWS cohort) was slowly lowered to the appropriate depth and permitted to sit for 5 min. ~100 nL of AAV solution was injected over the course of ~15 minutes via the Nanoject III system (Drummond Scientific). After an additional delay of 12 min, the micropipette was slowly raised. Once the optic cannula was secured in the stereotaxic frame and lowered to the appropriate DV coordinate, it was secured with dental composite (Tetric Evoflow; Heraeus Kulzer) that was cured with blue light. The HHMI headpost was delicately secured on the posterior area of the charisma ring with cyanoacrylate and dental composite. Finally, a single layer of dental composite was used to build and secure the rest of the headcap before being cured four times with blue light for 20 s each. The scalp was closed around the headcap by using cyanoacrylate.

Immediately after the optical implant and viral injection surgery, mice were placed in a sterile, clean cage that was half-on/half-off a heating pad until ambulation was observed. Mice were diligently monitored for 72 hr post-surgery. After this monitoring period, mice were transferred to a clean cage on a ventilated rack for at least 3 weeks prior to handling and water restriction.

## Handling and water restriction

After a 3-week waiting period, mice were placed under citric acid water restriction (*Urai et al., 2021*) during which mice were handled twice daily for 1 week. The bitterness of citric acid naturally causes mice to reduce their water consumption and, consequently, their weight while still having access to water. Initially, mice were acclimated to handling as researchers placed their hands into the cage for increasing amounts of time (e.g. 5 min to 10 min to 15 min). After mice became comfortable, they were held for increasing amounts of time (e.g. 2 min to 5 min to 10 min). Additionally, mice were acclimated to head fixation by holding their HHMI headposts (e.g. 30 s to 1 min to 2 min), and they were allowed to freely explore the behavioral tube for 5 min per handling session. Finally, mice were headfixed for increasing amounts of time in the behavioral setup (e.g. 5 min to 10 min to 15 min). Water was provided via transfer pipette to comfortably acclimate mice to head fixation. The head-fix apparatus contained a tube (length = 15 cm; inner diameter = 4 cm) that was affixed to a custom metal platform (length = 17 cm; width = 12 cm). The platform also contained HHMI mounting arms and holders for head fixation (*Huber et al., 2012*). During the final day of handling, mice were transitioned

to full water restriction as this permits greater control over motivation level. During behavioral testing, daily water intake was limited to ~1.5 mL per day to motivate performance on the Go/NoGo paradigm described below. The baseline body weight was measured daily during water restriction, and overall body weight was not permitted to drop below 80% of the baseline weight, consistent with levels of restriction used to motivate performance (*Guo et al., 2014*).

## Fiber photometry setup

Fiber photometry data were collected using a RZ10x lock-in amplifier within the Synapse suite (Tucker-Davis Technologies). This amplifier and accompanying Synapse software was used to control a custom-built optical benchtop through drivers (LEDD1B, Thorlabs) to modulate LED signals. Briefly, this optical benchtop consisted of a self-contained system of four 30 mm cage cubes with integrated filter mounts (CM1-DCH/M, Thorlabs). A 405 nm LED (M405L4, Thorlabs) and a 470 nm LED (M470L3, Thorlabs) were mounted onto the first 30 mm cage cube. The 405 nm LED was used to extract the calcium-independent isosbestic signal, and the 470 nm LED was used to acquire calcium-dependent axonal GCaMP signals during the Go/NoGo paradigm. The 405 nm isosbestic signal was modulated at 210 Hz, and the 470 nm GCaMP signal was modulated at 330 Hz. A 425 nm dichroic longpass filter (DMLP425R, Thorlabs) in the first cage cube reflected the 405 nm excitation light and permitted the 470 nm light to pass through. As the light entered the second cage cube, both excitation lights were reflected by a 495 nm dichroic longpass filter (495DCLP, 67–079, Edmund Optics) into the third cage cube. A 460/545 nm bandpass filter (69013xv2, Chroma) reflected both excitation wavelengths down to the subject via a low autofluorescence patch cable (MAF3L1, core = 400 μm, nA = 0.50, length = 1 m, Thorlabs). This cable was attached onto the implanted optical cannula (see above) by a ceramic mating sleeve (Thorlabs). Isosbestic and axon-jGCaMP8s emissions were collected via the optic cannula and passed through the 460/545 nm bandpass filter (69013xv2, Chroma)into the fourth. Finally, the emission fluorescence passed through the detection pathway to reach the RZ10x photosensors for online observation.

## Orofacial video recording

POm activity has been well correlated with whisking and pupil activity (*Petty et al., 2021*; *La Terra et al., 2022*). To analyze these dynamics, an LED driver controlled an IR spotlight that illuminated the contralateral eye and mystacial pad, and facial recordings were captured through an autofocusing USB webcam (NexiGo N660P) at ~20 fps within the Synapse software. To limit light pollution from outside sources (e.g. LED illumination within the brain/eye), an IR filter was placed in front of the webcam. Also, a shortpass emission filter at 750 nm (Chroma #ET750sp-2p8) was placed between the third and fourth cage cubes to prevenl recorded IR light from overloading the photosensors.

## Synchronization of task-related components

The Synapse software suite (Tucker-Davis Technologies) permitted the synchronous recording of both emitted (isosbestic and calcium) signals and video frames. Furthermore, the LabVIEW system-controlled paradigm-related components (e.g. texture movement, lick thresholds, trial type, trial outcome, etc.) and recorded the resulting behavioral parameters (e.g. licking activity, trial type, and trial outcome). To synchronize these two data streams for post-hoc analysis, TTL pulses relating to the texture arrival at its endpoint (which indicates the start of the presentation time window) from the LabVIEW system were captured within the Synapse software.

## In vivo optogenetics

For the JAWS cohort, a high-powered 617 nm LED (M617F2, Thorlabs) and current driver (LEDD1B, Thorlabs) were used for photoinactivation of POm axons in the striatum (*Huda et al., 2020*; *Nguyen et al., 2021*; *Mo et al., 2023*; *Chuong et al., 2014*). LED stimulation was provided at a probability of 0.50 for every session from the start of the Learning phase until mice reached the Expert phase (*La Terra et al., 2022*). The light intensity was measured at ~7 mW at the tip of the fiber. Stimulation intensity was kept consistent between mice and days by measuring the intensity with an optical power meter (PMD-100D, Thorlabs) prior to the first session every day. Light was delivered to the thalamic afferents in pDLS through an optical fiber patchcord (M95L01, fiber core diameter: 200 μm, length: 1 m, nA: 0.50) connected to a 200 μm core optical cannula (described above) via a mating sleeve

(Thorlabs). A small piece of black heat shrink tubing (Qualtek) was placed over the cannula during LED testing to prevent stray light from illuminating the presented texture during the task. The LabVIEW system controlled a Pulse Pal (*Sanders and Kepecs, 2014*) that activated the LED current driver to provide constant illumination for two seconds, evenly split 1 s before and 1 s after texture presentation, corresponding to the increased calcium activity observed during fiber photometry recordings.

For the No Stim cohort, a high-powered 470 nm LED (Prizmatix) and current driver (Prizmatix) were used for optogenetic activation. Note that no LED stimulation was provided during the Learning or Expert phase. Testing occurred during the first four sessions after the five initial Shaping sessions and the last four sessions after mice reached Expert status. Sessions consisted of 50 baseline trials followed by 10 alternating blocks of 10 OFF and 10 ON trials. The light intensity was measured at ~5 mW at the tip of the fiber and was kept consistent between mice and days by measuring the intensity with an optical power meter (PMD-100D, Thorlabs) prior to the first session on testing days. Light was delivered to the thalamic afferents in pDLS through an optical fiber patchcord (M73L01, fiber core diameter: 200 μm, length: 1 m, nA: 0.50, Thorlabs) connected to a 200 μm optical cannula (described above) via a mating sleeve (Thorlabs). A small piece of black heat shrink tubing (Qualtek) was placed over the cannula during LED testing to prevent stray light from illuminating the presented texture during the task. The LabVIEW system controlled a Pulse Pal (*Sanders and Kepecs, 2014*) that activated the LED current driver to provide pulsed illumination for 2 s at 15 Hz (matching the electrophysiology train photostimulation paradigm), evenly split 1 s before and 1 s after texture presentation.

## Go/NoGo whisker-based discrimination paradigm

Headfixed, water restricted mice were trained to perform a whisker-based discrimination paradigm as two textures were presented unilaterally to the right whiskers in a randomized order based on a custom-written LabVIEW code (National instruments). This code used transistor-transistor logic (TTL) pulses to control all aspects of the paradigm including a water delivery spout connected to a piezo film sensor (MSP1006-ND; Measurement Specialties), and a motorized linear stage (T-LSM100A; Zaber) with a stepper motor (T-NM17A04; Zaber) containing two windmill arms holding two different sandpaper textures (Go texture = 100 grit sandpaper, P100; NoGo texture = 1200 grit sandpaper, P1200; 3 M) as previously described (*Lee et al., 2019*; *Lee and Margolis, 2016*; *Margolis et al., 2012*; *Chen et al., 2013b*; *Chen et al., 2015*). Mice were trained to discriminate between these two textures by licking the piezo film sensor spout. Mechanical spout displacement resulted in transient voltage changes, and a lick was defined as voltage changes crossing either an upper or lower threshold once. After a lick was detected within the appropriate response window, the LabVIEW software immediately delivered the appropriate output: for Go trials, mice received a small water reward via a solenoid valve (0127; Buerkert) through the piezo spout, and, for NoGo trials, mice received a timeout period with co-occurring white noise. This paradigm occurred within a darkened room to minimize non-tactile related cues. Both textures and the headfix apparatus were cleaned with 70% ethanol between each mouse. If mice were not performing the task, sessions could be ended early. Water could be automatically delivered (AutoReward or AR) by the experimenter following 20 consecutive trials without a response. Finally, a session could be terminated if a mouse did not lick when water was present on the end of the spout after three ARs.

Trials began with a 1000ms pre-task interval followed by a brief cue tone (100ms, 2039 Hz) that accompanied windmill texture movement toward the mice. Once the windmill texture had reached a predetermined distance within the whisking field, mice had to respond within a 2000ms presentation time (PT) window. For the first 500ms of the PT window, a grace period was present where responses did not trigger appropriate outcomes to reduce impulsivity. If mice licked in response to the Go texture, the trial was considered a Hit and resulted in a water reward accompanied by a correct tone (2793 Hz). If mice licked in response to the NoGo texture, the trial was considered a False Alarm (FA) and resulted in punishment parameters including a time-out period (12,000ms) and an accompanying white noise during the time-out period. Air puffs were eschewed as a punishment parameter to permit continuous pupil dynamic recording. If mice did not lick to the Go or the NoGo texture, nothing occurred, and the trial was considered a Miss or a Correct Rejection (CR), respectively. Immediately after a lick was recorded or the 2000ms PT window had elapsed, the windmill texture retreated to its original position where the current texture either remained or the other texture was rotated into position. Finally, trials were separated by a 2000ms intertrial interval. Behavioral performance was

tracked across texture discrimination sessions by computing multiple behaviorally-related parameters including Hit Rate, FA Rate, Sensitivity (d') and Bias (*McNicol, 1970*). For the fiber photometry cohort, the 405 nm and 470 nm LEDs provided constant illumination throughout all trials. For the JAWS cohort, LED photoinactivation occurred at a trial probability of 0.50 for every session from the start of the Learning until the end of the Expert phase (*La Terra et al., 2022*). For the No Stim cohort, LED stimulation did not occur during training (i.e. during either the Learning or Expert phases). It only occurred for four sessions after the initial five shaping sessions, and four sessions after mice reached expert status.

This whisker-based discrimination paradigm lasted up to 3 weeks. The FP cohort were only tested once per day to limit photobleaching. The JAWS and No Stim cohorts were tested twice daily. Each session consisted of 150 trials. All cohorts progressed through three behavioral phases (Shaping, Learning, and Expert) that were segmented into five analytical time points (Shaping, Early Learning, Late Learning, and Expert). The Shaping time point was the same for all mice. For the first three sessions, mice were acclimated to reliably trigger water delivery by licking the water spout. During these sessions, neither texture was presented to the whiskers. After, mice proceeded to texture discrimination training still under the Shaping phase. For the last two sessions, both textures were presented simultaneously with a Go texture probability of 0.90 and 0.75, respectively. Go and NoGo trials were interleaved in a pseudorandom order determined by the LabVIEW software. After the 0.75 probability session, mice progressed into the Learning phase. For all following sessions, the Go texture probability was set to 0.50 with a maximum of three consecutive presentations of the same texture. Discrete behavioral time points were determined as follows. The Early Learning time point was considered the first two sessions of the Learning phase. The Late Learning time point was considered the last two sessions of the Learning phase before achieving expert status. This occurred when mice had a Hit Rate ≥0.80 and a FA Rate ≤0.30 for two consecutive sessions. A strict sensitivity threshold was not used due to artificially increased sensitivity (discrimination) values as Hit Rate and/or FA Rate approach their extremes (e.g. see FPOm-18 sessions 8–15 in *Figure 4—figure supplement 1*). After discrimination training (i.e. achieving expert status), mice in the fiber photometry cohort were subjected to a single Reward session to assess calcium, licking, and pupil activity in the absence of texture input and licking-related outcomes. During this session, the Go trial probability remained at 0.50, and the Zaber motor moved the windmill texture holder toward the whisker field, but the textures were rotated out of whisker range. Further, the upper and lower thresholds were set so that licking could not trigger outcomes before the end of the PT window. A water reward was automatically delivered at the end of the PT window during Go trials only. A whisker trim control session was performed in a subset of mice to confirm that mice used their whiskers to discriminate as previously observed (*Lee et al., 2019*).

## Histology

For electrophysiology experiments, the tissue block containing the injection site was stored overnight in 10% neutral-buffered formalin at 4 °C. After, it was transferred into 0.2% PBS Azide at 4 °C until sectioning. The tissue block was mounted onto a stage and sectioned in 0.1 M PBS at a thickness of 100 µm using a VT-1000 vibratome (Leica). Slices were mounted onto microscope slides using DAPI Fluoromount-G (Southern Biotech #0100–20) and coverslipped before confocal imaging.

Following all behavioral experiments, mice were anesthetized with an intraperitoneal injection of Ketamine-Xylazine (120 mg/kg Ketamine, 24 mg/kg Xylazine) and transcardially perfused with PBS followed by 10% neutral-buffered formalin. The brain was delicately extricated and stored in 10% neutral-buffered formalin overnight at 4 °C. Tissue was mounted onto a stage and sectioned at 100 µm using a VT-1000 vibratome (Leica). Sections were mounted and coverslipped using DAPI Fluoromount-G (Southern Biotech #0100–20). Confocal images were acquired using a LSM800 confocal laser scanning microscope (Zeiss) for injection site location verification, cannula placement, and viral expression in POm axons within pDLS and stereotypical POm-cortical projections in S1 L1 and L5a of all experimental mice (*Bureau et al., 2006*; *Zhang and Bruno, 2019*; *La Terra et al., 2022*; *Meyer et al., 2010*; *Ohno et al., 2012*). All data were acquired using the Zen software suite (Zeiss).

## Quantification and statistical analysis

### Behavioral responding analysis

Behavioral performance was tracked across texture discrimination sessions by computing multiple behaviorally-related parameters including Hit Rate, FA Rate, Sensitivity (d') and Bias (*McNicol, 1970*). Hit Rate was calculated as follows: [Hit / (Hit +Miss)], where hit is the number of correct Go trials and Miss is the number of incorrect Go trials. The FA Rate formula is similar to the Hit Rate formula except it replaces Hit with FA and Miss with CR: [FA / (FA +CR)]. Sensitivity illustrates the ability to discriminate between the signal (Go texture) and the noise (NoGo texture), and it is derived from signal detection theory (*McNicol, 1970*). It is calculated as follows: [normalized inverse–(Hit Rate) - normalized inverse (FA Rate)]. Finally, Bias illustrates the overall responding bias, independent of trial type. It is calculated as follows: [0.5 * normalized inverse (Hit Rate)+normalized inverse (FA Rate)].

### Pupil analysis

After all data were captured, the recorded orofacial videos were analyzed using DeepLabCut, a deep learning model for pose estimation, to estimate pupil dynamics as mice learned to discriminate between the two textures (*Mathis et al., 2018*; *Nath et al., 2019*). Briefly, a model (*Yamada and Toda, 2022*) was created with eight markers circumscribing the pupil, permitting the estimation of the approximate pupil area for each frame. All videos were cropped to a smaller dimension (230x275 pixels) that focused on each mouse's face. For each mouse, one video was selected at varying behavioral time points, and 30 frames were extracted and manually labeled with the eight markers. Each marker corresponded to a pupil location: top, top right, right, bottom right, bottom, bottom left, left, and top left. Additionally, another marker was placed on a static location (e.g. the water spout) to label frames when blinking occurred. Overall, the initial training dataset contained 150 labeled frames, and the model was trained on this dataset with a ResNet50-based neural network for 250,000 iterations. After, the initial training videos were analyzed to assess the performance of the model from its marker estimations. 25 outlier frames from the initial training videos were extracted, manually corrected, and merged with the initial dataset. The model was trained again on this 275 labeled frame dataset with a ResNet50-based neural network for 250,000 iterations. After evaluating the network, the train error was calculated at 1.24 pixels, and the test error was measured at 1.25 pixels. Once the model was successfully trained, all behavioral videos of the five mice were analyzed. To estimate pupil area, a python library (scikit-image; *van der Walt et al., 2014*) was used to fit an ellipse on the estimations of the eight markers generated by DeepLabCut. This ellipse model was used to predict the values of the vertices and co-vertices on the ellipse, permitting the calculation of the major and minor axes. Thus, these measurements were used to approximate the area of the pupil for each frame. To detect blinking behavior, the maximum pupil size in non-blinking conditions was calculated and used as a threshold to identify abnormally high pupil predictions. As such, pupil areas greater than 450 pixels were removed via interpolation. The pupil area data were saved as a csv file for importing into MATLAB.

### Fiber photometry signal processing

Custom-written MATLAB scripts were used for post processing of the fluorescent signals. Photobleaching was corrected in both the isosbestic and GCaMP signals using detrended lines of best fit and subtracting the line from all values. After, the isosbestic and GCaMP median absolute deviation of z-score (ZMAD) signals were calculated before subtracting the GCaMP signal from the isosbestic to remove calcium-independent artifacts.

### Alignment of task-related components

A custom-written MATLAB script imported the lick-related and overall responding data, the processed ZMAD GCaMP signal, and the processed pupil csv for alignment. The overall responding data (containing trial elements such as time of PT window start which is the TTL flag within Synapse) was converted from UNIX timecode into seconds to match the Synapse (containing the pupil video and processed GCaMP signal) time. Pupil data was resampled to align with GCaMP signal using rational fraction approximation. Furthermore, the length of each trial, as determined by the overall responding data, was recorded and used to capture the GCaMP and pupil data within each trial window. Licks were identified by setting upper and lower thresholds, and detecting when either was crossed. At this

point, all parameters (e.g. GCaMP, pupil, and licking activity) were captured and aligned within the time window of each trial. These parameters could now be segmented by trial type (e.g. Go vs. NoGo texture) and by trial outcome (e.g. Hit, Miss, FA, CR) for more advanced analysis.

## Calcium analysis

auROC is an analysis commonly applied to calcium imaging data to characterize the stereotypy of neuronal responses (*Li et al., 2017*; *Kingsbury et al., 2019*). Briefly, a baseline window is set within a non-task-related component of the overall calcium signal and compared to the rest of the signal via a sliding window. The maximal value of each signal is captured at each behavioral time point and averaged across the cohort. auROC values equal to 0.50 indicate no differences between the baseline signal and the task-related signal.

To assess longitudinal changes in calcium activity, two windows were established: a Control Window located from trial start to the sound cue indicating trial start, and a Target Window located 2 s before and after PT start (overall = 4 s). For each trial, all calcium peaks were measured, and only peaks ≥90 th percentile with a minimum peak prominence of 2 were selected. Finally, the maximum values within the Control and Target Windows (if present) were captured and averaged for every session (*Legaria et al., 2022*).

## Statistical analyses

All data are reported as mean ± SEM unless otherwise noted. Statistical analyses were performed in GraphPad Prism (USA). All data were tested using the Shapiro-Wilk normality test. The means of different data distributions were analyzed and compared using two-tailed Student's t-test (*Figure 1—figure supplement 1* panels O and R; *Figure 2E* FA Rate, 2E Bias, 3 F, 4E Hit Rate, 4E Sensitivity, 4E Bias, 4 G FA Rate, 4 G Bias, 4 K Learning/Expert Hit Rate, 4 K Learning/Expert FA Rate, 4 K Learning/ Expert Sensitivity, 4 K Learning/Expert Bias, 4 L; *Figure 4—figure supplement 1C*, Hit Rate, FA Rate, Sensitivity, Bias), Wilcoxon signed rank test (*Figure 1—figure supplement 1N and Q*; *Figure 2E* Hit Rate, 2E Sensitivity, 4E FA Rate, 4 G Hit Rate, 4 G Sensitivity), ordinary one-way ANOVA with Tukey's multiple comparison correction (*Figure 1—figure supplement 1G and L*), Kruksal-Wallis with Dunn's multiple comparison correction (*Figures 1E, F, I, L, H, I, J, K and 4I*), linear regression (*Figure 1—figure supplement 1E*), Repeated measures one-way ANOVA with Tukey's multiple comparison correction (*Figure 3B and C*), Repeated measures mixed-effects analysis with Tukey's multiple comparison correction (*Figure 3—figure supplement 1D, top and bottom*), Repeated measures two-way ANOVA with Sidak's multiple comparison correction (*Figure 3K and L*), Repeated measures mixed-effects analysis with Sidak's multiple comparison correction (*Figure 3Q*). For all statistical tests, *p<0.05, **p<0.01, ***p<0.001, and ****p<0.0001.

## Acknowledgements

The authors would like to thank all members of the Margolis lab for their comments related to manuscript text and figures. This work was generously supported by funding from the National Institutes of Health (F31NS117093, AJY); (NCATS TL1TR003019, AJY); (R01NS094450, DJM), the National Science Foundation (IOS-1845355, DJM), and the Rutgers Busch Biomedical Grant Program (IL/DJM).

## Additional information

### Funding

| Funder | Grant reference number | Author |
| --- | --- | --- |
| National Institutes of Health | F31NS117093 | Alex J Yonk |
| National Science Foundation | IOS-1845355 | David J Margolis |
| National Institutes of Health | R01NS094450 | David J Margolis |

| Funder | Grant reference number | Author |
| --- | --- | --- |
| National Institutes of Health | NCATS TL1TR003019 | Alex J Yonk |
| Rutgers Busch Biomedical | | Ivan Linares-García<br>David J Margolis |

The funders had no role in study design, data collection and interpretation, or the decision to submit the work for publication.

### Author contributions

Alex J Yonk, Conceptualization, Resources, Data curation, Software, Formal analysis, Supervision, Funding acquisition, Validation, Investigation, Visualization, Methodology, Writing – original draft, Project administration, Writing – review and editing; Ivan Linares-García, Resources, Software, Investigation, Methodology; Logan Pasternak, Software, Methodology; Sofia E Juliani, Software, Formal analysis, Methodology; Mark A Gradwell, Arlene J George, Supervision, Methodology; David J Margolis, Conceptualization, Supervision, Funding acquisition, Writing – original draft, Project administration, Writing – review and editing

### Author ORCIDs

Alex J Yonk https://orcid.org/0000-0002-1366-889X
Ivan Linares-García https://orcid.org/0000-0002-0622-4916
David J Margolis https://orcid.org/0000-0002-2678-4216

### Ethics

This study was performed in accordance with the recommendations in the Guide for the Care and Use of Laboratory Animals of the National Institutes of Health. Animals were handled according to approved institutional animal care and use committee (IACUC) protocols (#999900197) of Rutgers, The State University of New Jersey. Every effort was made to minimize suffering.

Reviewer #1 (Public review): https://doi.org/10.7554/eLife.98563.3.sa1
Reviewer #2 (Public review): https://doi.org/10.7554/eLife.98563.3.sa2
Reviewer #3 (Public review): https://doi.org/10.7554/eLife.98563.3.sa3
Author response https://doi.org/10.7554/eLife.98563.3.sa4

# Additional files

### Supplementary files

MDAR checklist

### Data availability

Data are available at https://github.com/margolislab/Yonk_2024/ (copy archived at *margolislab, 2025*).

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
