## [Editor Report · eLife Assessment]

Yonk and colleagues provide a **valuable**, timely, and in-depth study showcasing the role of thalamostriatal inputs in learning and action selection. After characterizing the synaptic properties of these inputs onto different striatal cell types in vitro, they provide **solid** evidence that posterior medial thalamic nucleus (POm) terminals in striatum are activated during reward expectation and arousal. The overall function of this pathway and the degree to which results are confounded by viral contamination of surrounding nuclei and movements remain open questions.

---

## [Referee Report · Reviewer #1 (Public review)]

Summary:

This work aims at understanding the role of thalamus POm in dorsal lateral striatum (DLS) projection in learning a sensorimotor associative task. The authors first confirm that POm forms "en passant" synapses with some of the DLS neuronal subtypes. They then perform a go/no-go associative task that consists of the mouse learning to discriminate between two different textures and to associate one of them with an action. During this task they either record the activity of the POm to DLS axons using endoscopy or silence their activity. They report that POm axons in the DLS are activated around the sensory stimulus but that the activity is not modulated by the reward. Last, they showed that silencing the POm axons at the level of DLS slows down learning the task.

The authors show convincing evidence of projections from POm to DLS and that POm inputs to DLS code for whisking whatever the outcome of the task is. However, their results do not allow to conclude if more neurones are recruited during the learning process or if the already activated fibres get activated more strongly. Last, because POm fibres in the DLS are also projecting to S1, silencing the POm fibres in the DLS could have affected inputs in S1 as well and therefore, the slowdown in acquiring the task is not necessarily specific to the POm to DLS pathway.

Strengths:

One of the main strengths of the paper is to go from slice electrophysiology to behaviour to get an in-depth characterization of one pathway. The authors did a comprehensive description of the POm projections to the DLS using transgenic mice to unambiguously identify the DLS neuronal population. They also used a carefully designed sensorimotor association task, and they exploited the results in depth.

It is a very nice effort to have measured the activity of the axons in the DLS not only after the mice have learned the task but throughout the learning process. It shows the progressive increase of activity of POm axons in the DLS, which could imply that there is a progressive strengthening of the pathway. The results show convincingly that POm axons in the DLS are not activated by the outcome of the task but by the whisker activity, and that this activity in average increases with learning.

Weaknesses:

One of the main targets of the striatum from thalamic input are the cholinergic neurons that weren't investigated here, is there information that could be provided?

It is interesting to know that the POm projects to all neuronal types in the DLS, but this information is not used further down the manuscript so the only take-home message of Figure 1 is that the axons that they image or silence in the DLS are indeed connected to DLS neurons and not just passing fibres. In this line, are these axons the same as the ones projecting to S1? If this is the case, why would we expect a different behaviour of the axon activity at the DLS level compared to S1?

The authors used endoscopy to measure the POm axons in the DLS activity, which makes it impossible to know if the progressive increase of POm response is due to an increase of activity from each individual neurons or if new neurons are progressively recruited in the process.

The picture presented in Figure 4 of the stimulation site is slightly concerning as there are hardly any fibres in neocortical layer 1 while there seems to be quite a lot of them in layer 4, suggesting that the animal here was injected in the VB. This is especially striking as the implantation and projection sites presented in Figure 1 and 2 are very clean and consistent with POm injection.

Comment after review: The weaknesses remain as concerns have not been addressed. The dataset is interesting but the interpretation, due partly to the lack of control (especially relative to VPM contamination), is difficult.

---

## [Referee Report · Reviewer #2 (Public review)]

Summary:

Yonk and colleagues show that the posterior medial thalamus (POm), which is interconnected with sensory and motor systems, projects directly to major categories of neurons in the striatum, including direct and indirect pathway MSNs, and PV interneurons. Activity in POm-striatal neurons during a sensory-based learning task indicates a relationship between reward expectation and arousal. Inhibition of these neurons slows reaction to stimuli and overall learning. This circuit is positioned to feed salient event activation to the striatum to set the stage for effective learning and action selection.

Strengths:

The results are well presented and offer interesting insight into an understudied thalamostriatal circuit. In general, this work is important as part of a general need for an increased understanding of thalamostriatal circuits in complex learning and action selection processes, which have generally received less attention than corticostriatal systems.

Weaknesses:

There could be a stronger connection between the connectivity part of the data - showing that POm neurons context D1, D2, and PV neurons in striatum but with some different properties - and the functional side of the project. One wonders whether the POm neurons projecting to these subtypes or striatal neurons have unique signaling properties related to learning, or if there is a uniform, bulk signal sent to striatum. This is not a weakness per se, as it's reasonable for these questions to be answered in future papers.

All the in vivo activity-related conclusions stem from data from just 5 mice, which is a relatively small sample set. Optogenetic groups are also on the small side.

Comments on revisions:

The revision has a lot of thoughtful discussion added. I think overall the paper is more thorough and will also be a nice set up for a number of future research questions.

---

## [Referee Report · Reviewer #3 (Public review)]

Yonk and colleagues investigate the role of the thalamostriatal pathway. Specifically, they studied the interaction of the posterior thalamic nucleus (PO) and the dorsolateral striatum in the mouse. First, they characterize connectivity by recording DLS neurons in in vitro slices and optogenetically activating PO terminals. PO is observed to establish depressing synapses onto D1 and D2 spiny neurons as well as PV neurons. Second, the image PO axons are imaged by fiber photometry in mice trained to discriminate textures. Initially, no trial-locked activity is observed, but as the mice learn PO develops responses timed to the audio cue that marks the start of the trial and precedes touch. PO does appear to encode the tactile stimulus type or outcome. Optogenetic suppression of PO terminals in striatum slow task acquisition. The authors conclude that PO provides a "behaviorally relevant arousal-related signal" and that this signal "primes" striatal circuitry for sensory processing.

A great strength of this paper is its timeliness. Thalamostriatal processing has received almost no attention in the past, and the field has become very interested in the possible functions of PO. Additionally, the experiments exploit multiple cutting-edge techniques.

There seem to be some technical/analytical weaknesses. The in vitro experiments appear to have some contamination of nearby thalamic nuclei by the virus delivering the opsin, which could change the interpretation. Some of the statistical analysis of these data also appear inappropriate. The correlative analysis of Pom activity in vivo, licking, and pupil could be more convincingly done.

The bigger weakness is conceptual - why should striatal circuitry need "priming" by thalamus in order to process sensory stimuli? Why would such circuitry even be necessary? Why is a sensory signal from cortex insufficient? Why should the animal more slowly learn the task? How does this fit with existing ideas of striatal plasticity? It is unclear from the experiments that the thalamostriatal pathway exists for priming sensory processing. In fact the optogenetic suppression of the thalamostriatal pathway seems to speak against that idea.

Comments on revisions:

The authors have only tweaked the Discussion and not necessarily in ways that addressed our previous comments. They could have fairly easily analyzed the effect of distance of recording from injection site and compared subsets of data depending on contamination beyond PO (my comments 1 and 2) or effects of movements (3 and 4). Minimally, they could have given caveats in the Results and Discussion about these, and I would strongly encourage them to be explicit about the caveats. The analyses would probably be better.

The suggestion that the effects have something to do with priming (5), seems a grasp for function of the circuit.

---

## [Author Response]

The following is the authors’ response to the original reviews.

**Public Reviews:**

**Reviewer #1 (Public Review):**
Summary:This work aims to understand the role of thalamus POm in dorsal lateral striatum (DLS) projection in learning a sensorimotor associative task. The authors first confirm that POm forms "en passant" synapses with some of the DLS neuronal subtypes. They then perform a go/no-go associative task that consists of the mouse learning to discriminate between two different textures and to associate one of them with an action. During this task, they either record the activity of the POm to DLS axons using endoscopy or silence their activity. They report that POm axons in the DLS are activated around the sensory stimulus but that the activity is not modulated by the reward. Last, they showed that silencing the POm axons at the level of DLS slows down learning the task.The authors show convincing evidence of projections from POm to DLS and that POm inputs to DLS code for whisking whatever the outcome of the task is. However, their results do not allow us to conclude if more neurons are recruited during the learning process or if the already activated fibres get activated more strongly. Last, because POm fibres in the DLS are also projecting to S1, silencing the POm fibres in the DLS could have affected inputs in S1 as well and therefore, the slowdown in acquiring the task is not necessarily specific to the POm to DLS pathway.

We thank the reviewer for these constructive comments. The points are addressed below.

Strengths:One of the main strengths of the paper is to go from slice electrophysiology to behaviour to get an in-depth characterization of one pathway. The authors did a comprehensive description of the POm projections to the DLS using transgenic mice to unambiguously identify the DLS neuronal population. They also used a carefully designed sensorimotor association task, and they exploited the results in depth.It is a very nice effort to have measured the activity of the axons in the DLS not only after the mice have learned the task but throughout the learning process. It shows the progressive increase of activity of POm axons in the DLS, which could imply that there is a progressive strengthening of the pathway. The results show convincingly that POm axons in the DLS are not activated by the outcome of the task but by the whisker activity, and that this activity on average increases with learning.Weaknesses:One of the main targets of the striatum from thalamic input are the cholinergic neurons that weren't investigated here, is there information that could be provided?

This is true of the parafascicular (Pf) thalamic nucleus, which has been well studied in this context. However, there is much less known about the striatal projections of other thalamic nuclei, including POm, and their inputs to cholinergic neurons. Anatomical tracing evidence from Klug et al. (2018), which mapped brain-wide inputs to striatal cholinergic (ChAT) interneurons, suggests that Pf provides the majority of thalamic innervation of striatal ChAT neurons compared to other thalamic nuclei. Many other thalamic nuclei, including POm, showed very little of no labeling, suggesting weak innervation of ChAT interneurons. However, it is possible that these thalamic nuclei, including POm, do provide functional innervation of ChAT interneurons that is not sufficiently assessed by anatomical tracing. Understanding the innervation patterns of POm-striatal projections beyond the three cell types we have studied here would be an important area of further study.

It is interesting to know that the POm projects to all neuronal types in the DLS, but this information is not used further down the manuscript so the only take-home message of Figure 1 is that the axons that they image or silence in the DLS are indeed connected to DLS neurons and not just passing fibres. In this line, are these axons the same as the ones projecting to S1? If this is the case, why would we expect a different behaviour of the axon activity at the DLS level compared to S1?

Tracing of single POm axons by Ohno et al. (2012) indicated that POm axons form a branched collateral that innervates striatum, while the main axon continues in the rostral-dorsal direction to innervate cortex. We think it is reasonable, based on the morphology, that our optogenetic suppression experiment restricted the suppression of glutamate release to this branch and avoided the other branches of the axon that project to cortex. However, testing this would require monitoring S1 activity during the POm-striatal axon suppression, which we did not do in this study.

It is a very interesting question whether there could be different axon activity behavior in striatum versus S1. There is surprising evidence that POm synaptic terminals are different sizes in S1 and M1 and show different synaptic physiological properties depending on these cortical projection targets (Casas-Torremocha et al., 2022). Based on this, it is possible that POm-striatal synapses show distinct properties compared to cortex; however, this will need to be tested in future work.

The authors used endoscopy to measure the POm axons in the DLS activity, which makes it impossible to know if the progressive increase of POm response is due to an increase of activity from each individual neuron or if new neurons are progressively recruited in the process.

This is a good point. It would be necessary to perform chronic two-photon imaging of POm neurons (or chronic electrophysiological recordings) to determine whether the activity of individual neurons increased versus whether individual neuron activity levels remained similar but new neurons became active with learning. Even under baseline conditions, it is not known in detail what fraction of the population of POm neurons is active during sensory processing or behavior, highlighting how much is still to be discovered in this exciting area of neuroscience.

The picture presented in Figure 4 of the stimulation site is slightly concerning as there are hardly any fibres in neocortical layer 1 while there seems to be quite a lot of them in layer 4, suggesting that the animal here was injected in the VB. This is especially striking as the implantation and projection sites presented in Figures 1 and 2 are very clean and consistent with POm injection.

Although this image was selected to demonstrate the position of the POm injection site and optical fiber implant above striatal axons, the reviewer is correct that there appears to be mixed labeling of axons in L4 and L5a. In some cases, there was expression slightly outside the border of POm (see Fig. 1B, right), which might explain the cortical innervation pattern in this figure. While cortically bound VPM axons pass through the striatum, they do not form synaptic terminals until reaching the cortex (Hunnicutt et al., 2016). If, as may be the case, inhibitory opsins suppress release of neurotransmitter at synaptic terminals more effectively than action potential propagation in axons, it may be likely that optogenetic suppression of POm-striatal terminals is more effective than suppression of action potentials in off-target-labelled VPM axons of passage. Ideally, we could compare effects of suppression of POm-striatal synapses with POm-cortical synapses and VPM-cortical synapses, but this was outside the bandwidth of the present study.

**Reviewer #2 (Public Review):**
Summary:Yonk and colleagues show that the posterior medial thalamus (POm), which is interconnected with sensory and motor systems, projects directly to major categories of neurons in the striatum, including direct and indirect pathway MSNs, and PV interneurons. Activity in POm-striatal neurons during a sensory-based learning task indicates a relationship between reward expectation and arousal. Inhibition of these neurons slows reaction to stimuli and overall learning. This circuit is positioned to feed salient event activation to the striatum to set the stage for effective learning and action selection.Strengths:The results are well presented and offer interesting insight into an understudied thalamostriatal circuit. In general, this work is important as part of a general need for an increased understanding of thalamostriatal circuits in complex learning and action selection processes, which have generally received less attention than corticostriatal systems.Weaknesses:There could be a stronger connection between the connectivity part of the data - showing that POm neurons context D1, D2, and PV neurons in the striatum but with some different properties - and the functional side of the project. One wonders whether the POm neurons projecting to these subtypes or striatal neurons have unique signaling properties related to learning, or if there is a uniform, bulk signal sent to the striatum. This is not a weakness per se, as it's reasonable for these questions to be answered in future papers.

We are very interested to understand the potentially distinct learning-related synaptic and circuit changes that potentially occur at the POm synapses with D1- and D2-SPNs and PV interneurons, and other striatal cell types. We agree that this would be an important topic for further investigation.

All the in vivo activity-related conclusions stem from data from just 5 mice, which is a relatively small sample set. Optogenetic groups are also on the small side.

We appreciate this point and agree that higher N can be important for observing robust effects. A factor of our experiments that helped reduce the number of animals used was the longitudinal design, with repeated measures in the same subjects. This allowed for the internal control of comparing learning effects in the same subject from naïve to expert stages and therefore increased robustness. Even with relatively small group sizes, results were statistically significant, suggesting that the use of more mice was unnecessary, which we considered consistent with best practice in the use of animals in research. We also note that our group sizes were consistent with other studies in the field.

**Reviewer #3 (Public Review):**
Yonk and colleagues investigate the role of the thalamostriatal pathway. Specifically, they studied the interaction of the posterior thalamic nucleus (PO) and the dorsolateral striatum in the mouse. First, they characterize connectivity by recording DLS neurons in in-vitro slices and optogenetically activating PO terminals. PO is observed to establish depressing synapses onto D1 and D2 spiny neurons as well as PV neurons. Second, the image PO axons are imaged by fiber photometry in mice trained to discriminate textures. Initially, no trial-locked activity is observed, but as the mice learn PO develops responses timed to the audio cue that marks the start of the trial and precedes touch. PO does appear to encode the tactile stimulus type or outcome. Optogenetic suppression of PO terminals in striatum slow task acquisition. The authors conclude that PO provides a "behaviorally relevant arousal-related signal" and that this signal "primes" striatal circuitry for sensory processing.A great strength of this paper is its timeliness. Thalamostriatal processing has received almost no attention in the past, and the field has become very interested in the possible functions of PO. Additionally, the experiments exploit multiple cutting-edge techniques.There seem to be some technical/analytical weaknesses. The in vitro experiments appear to have some contamination of nearby thalamic nuclei by the virus delivering the opsin, which could change the interpretation. Some of the statistical analyses of these data also appear inappropriate. The correlative analysis of Pom activity in vivo, licking, and pupil could be more convincingly done.The bigger weakness is conceptual - why should striatal circuitry need "priming" by the thalamus in order to process sensory stimuli? Why would such circuitry even be necessary? Why is a sensory signal from the cortex insufficient? Why should the animal more slowly learn the task? How does this fit with existing ideas of striatal plasticity? It is unclear from the experiments that the thalamostriatal pathway exists for priming sensory processing. In fact, the optogenetic suppression of the thalamostriatal pathway seems to speak against that idea.

We thank the reviewer for these constructive comments. The points are addressed below.

**Recommendations for the authors:**

**Reviewer #2 (Recommendations For The Authors):**
Do POm neurons innervate CINs also? The connection between the PF thalamus and CINs is mentioned in a couple of places - one question is how unique are the input patterns for the POm versus adjacent sensorimotor thalamic regions, including the PF? This isn't a weakness per se but knowing the answer to that question would help in forming a more complete picture of how these different thalamostriatal circuits do or do not contribute uniquely to learning and action selection.

Anatomical tracing evidence from Klug et al. (2018), which mapped brain-wide inputs to striatal cholinergic (ChAT) interneurons, suggests that Pf provides the majority of thalamic innervation of striatal ChAT neurons compared to other thalamic nuclei. Many other thalamic nuclei, including POm, showed very little or no labeling, suggesting weak innervation of ChAT interneurons. However, it is possible that these thalamic nuclei, including POm, do provide functional innervation of ChAT interneurons that is not sufficiently assessed by anatomical tracing.

Another difference between Pf and other thalamic nuclei (likely including POm) comes from anatomical tracing evidence (Smith et al., 2014; PMID: 24523677) which indicates that Pf inputs form the majority of their synapses onto dendritic shafts of SPNs, while other thalamic nuclei form synapses onto dendritic spines. Understanding the innervation patterns of POm-striatal projections beyond the three cell types we have studied here, including ChAT neurons and subcellular localization, would be an important area of further study.

It would be useful to know to what extent these POm-striatum neurons are activated generally during movement, versus this discrimination task specifically.

We agree that distinguishing general movement-related activity from task-specific activity would be very useful. Earlier work (Petty et al., 2021) showed a close relationship between POm neuron activity, spontaneous (task-free) whisker movements, and pupil-indexed arousal in head-restrained mice. Oram et al. (2024; PMID: 39003286) recently recorded VPM and POm in freely moving mice during natural movements, finding that activity of both nuclei correlated with head and whisker movements. These studies indicate that POm is generally coactive with exploratory head and whisker movements.

During task performance, the situation may change with training and attentional effects. For example, Petty and Bruno (2024) (https://elifesciences.org/reviewed-preprints/97188) showed that POm activity correlates more closely with task demands than tactile or visual stimulus modality. Our data indicate that POm axonal signals are increased at trial start during anticipation of tactile stimulus delivery and through the sensory discrimination period, then decrease to baseline levels during licking and water reward collection (Fig. 3). Results of Petty and Bruno (2024) together with ours suggest that POm is particularly active during the context of behaviorally relevant task performance. Thus, we think it is likely that, while pupil dilation indexes general movement and arousal, POm activity is more specific to movement and arousal associated with task engagement and behavioral performance. We have strengthened this point in the Discussion.

Many of the data panels and text for legends/axes are quite small, and the stroke on line art is quite faint - overall figures could be improved from a readability standpoint.

We thank the reviewer for their careful attention to the figures.

**Reviewer #3 (Recommendations For The Authors):**
Major(1) Page 4, the Results regarding PSP and distance from injection site. The r-squared is the wrong thing to look at to test for a relationship. One should look at the p-value on the coefficient corresponding to the slope. The p-value is probably significant given the figures, in which case there may be a relationship contrary to what is stated. All the low r-squared value says is that, if there is a relationship, it does not explain a lot of the PSP variability.

We thank the reviewer for alerting us this oversight. We have included the p value (p = 0.0293) in the figure and legend, and indicated that the relationship is “small but significant”.

(2) Figure 1B suggests that the virus injections extend beyond POm and into other thalamic structures. Do any of the results change if the injections contaminating other nuclei are excluded from the analysis? I am not suggesting the authors change the figures/analyses. I am simply suggesting they double-check.

We selected for injections that were predominantly expressing in POm as determined by post-hoc histological analysis (see Fig. 1, right). As above, we think that axons of passage that do not form striatal synapses are less likely to be suppressed than axons with terminals; however, this would need to be determined in further experiments. Because the preponderance of expression is within POm, we think the results would be similar even with a stricter selection criterion.

(3) The authors conclude that POm and licking are not correlated (bottom of page 6 pertaining to Figures 3A-F). The danger of these analyses is that they assume that GCaMP8 is a perfect linear reporter of POm spikes. The reliability of GCaMP8 has been quantified in some cell types, but not thalamic neurons, which have relatively higher firing rates.

The reviewer is correct that the relationship between GCaMP8 fluorescence changes and spiking has not been sufficiently characterized in thalamic neurons, and that this would be important to do.

What if the indicator is simply saturated late into the trial (after the average reaction time)? It would look like there is no response and one would conclude no correlation, but there could be a very strong correlation.

While saturation is worthy of concern, the signal dynamics here argue against this possibility. The reason is that the signal increased in the early part of the trial and decreased by the end. If saturation was an issue, this would have been apparent during the initial increase. When the signal decreased in amplitude at the end of the trial, this indicates that the signal is not saturated because it is returning from a point closer to its maximum (and is becoming less saturated).

Also, what happens between trials? Are the correlations the same, stronger, weaker? Ideally, the authors would analyze the data during and between trials.

Between trials the signal did not show further changes in baseline beyond what was displayed at the start and end of behavioral trials. There were no consistent increases or decreases in signals between trials, except perhaps during strong whisking bouts. This is anecdotal because we did not analyze between-trial data. However, it is interesting and important to note that signals increased dramatically in amplitude from naïve, early learning to expert behavioral performance (Fig. 3), highlighting that POm-axonal signals relate to behavioral engagement and performance rather than spontaneous behaviors.

(4) Axonal activity could also appear more correlated with the pupil than licking because pupil dynamics are slow like the dynamics of calcium indicators. These kernels could artificially inflate the correlation. Ideally, the authors could consider these temporal effects. Perhaps they could deconvolve the temporal profiles of calcium and pupil before correlating? Or equivalently incorporate the profiles into their analysis?

We analyzed the lick probability histograms, which had a temporal profile similar to the calcium signals (Fig. 3D,E), ruling out concerns about effects of temporal effects on correlations. It is also worth noting that we observed changes in correlations between calcium signals and pupil with learning stage (Fig. 3I), even though the temporal profiles (signal dynamics) are not changing. Thus, temporal effects of the signals themselves are not the driver of correlations, but rather the changes in relative timing between calcium signals and pupil, as occur with learning.

(5) The authors conclude that PO provides a "behaviorally relevant arousal-related signal" and that this signal "primes" striatal circuitry for sensory processing. The data here support the first part. It is not clear that the data support the second part, largely because it is vague what "priming" of sensory processing or "a key role in the initial stages of action selection (p.9) even means here. Why would such circuitry even be necessary? Why is a sensory signal from the cortex insufficient? Why should the animal more slowly learn the task? How does this fit with existing ideas of striatal plasticity? Some conceptual proposals from the authors, even if speculative and not offered as a conclusion, would be helpful.

We appreciate these good points and have added further consideration and revision of the concept of priming and potential roles in an extensively revised Discussion section.

(6) The photometry shows that PO turns on about 2 seconds before the texture presentation. PO's activity seems locked to the auditory cue, not the texture (Figure 2). This means that the attempt to suppress the thalamostriatal pathway with JAWS (Figure 4) is rather late, isn't it? Some PO signals surely go through. This seems to contradict the idea of priming above. It would be good if the authors could factor this into their narrative. Perhaps labelling the time of the auditory cue in Figure 4C would also be helpful.

The start of texture presentation (movement of the texture panel toward the mouse) and auditory cue occur at the same time. To clarify this, we added a label “start tone” in Figure 4C and also in Figure 2C.

For optogenetic (JAWS) suppression, we intentionally chose a time window between start tone onset and texture presentation, because our photometry experiments showed that this was when the preponderance of the signal occurred. However, the reviewer is correct that our chosen optogenetic suppression (JAWS) onset occurs shortly after the photometry signal has already started, potentially leaving the early photometry signal un-suppressed. Our motivation for choosing a restricted time window surrounding the texture presentation time was (1) to minimize illumination and potential heating of brain tissue; (2) to target a time window that avoids the auditory cue but covers stimulus presentation. We did not want to extend the duration of the suppression to before the trial started, because this could produce task-non-specific effects, such as distraction or loss of attention before the start of the trial.

Even if some signal were getting through before suppression, we don’t think this contradicts the possibility of ‘priming’, because the process underlying priming would still be disrupted even if not totally suppressed. This would alter the temporal relationship between POm-striatal inputs and further corticostriatal inputs (from S1 and M1 cortex, for example). We have included further consideration of these points and possible relation to the priming concept in the Discussion.

Minor(1) Page 5, "the sensitivity metric is artificially increased". What do you mean "artificially"? The mice are discriminating better. It is true that either a change in HR or FAR can cause the sensitivity metric to change, but there is nothing artificial or misleading about this.

We removed the word artificial and clarified our definition of behaviorally Expert in this context:

“Mice were considered Expert once they had reached ≥ 0.80 Hit Rate and ≤ 0.30 FA Rate for two consecutive sessions in lieu of a strict sensitivity (d’) threshold; we found this definition more intuitive because d’ is enhanced as Hit Rate and FA Rate approach their extremes (0 or 1)”

(2) Page 7, "Upon segmentation (Figure S4G-J)". Do you mean "segregation by trial outcome"?

Corrected.

(3) Page 9, "POm projections may have discrete target-specific functions, such that POm-striatal inputs may play a distinct role in sensorimotor behavior compared to POm-cortical inputs". Would POm-cortical inputs not also be sensorimotor? The somatosensory cortex contains a lot of corticostriatal cells. It also has various direct and indirect links to the motor cortex as well.

We have clarified the wording here to convey the possibility that POm signals could be received and processed differently by striatal versus cortical circuitry, and have moved this statement to later in the discussion for better elaboration.

(4) The Methods state that male and female mice were used. Why not say how many of each and whether or not there are any sex-specific differences?

We added the following information to the Methods:

The number of male and female mice were as follows, by experiment type: 6 male, 4 female (electrophysiology); 3 male, 2 female (fiber photometry); 4 male, 5 female (optogenetics). Data were not analyzed for sex differences.